# A general framework for predicting the transcriptomic consequences of non-coding variation and small molecules

**Moustafa Abdalla**[1,2,3,4]*, **Mohamed Abdalla**[5,6]*

**1** Wellcome Trust Centre for Human Genetics, Nuffield Department of Medicine, University of Oxford, Oxford, United Kingdom, **2** Oxford Centre for Diabetes, Endocrinology and Metabolism, Radcliffe Department of Medicine, University of Oxford, Oxford, United Kingdom, **3** Computational Statistics and Machine Learning, Department of Statistics, University of Oxford, Oxford, United Kingdom, **4** Department of Surgery, Harvard Medical School, Boston, Massachusetts, United States of America, **5** Vector Institute for Artificial Intelligence, Toronto, Canada, **6** Department of Computer Science, University of Toronto, Toronto, Canada

* moustafa_abdalla@hms.harvard.edu (MA); mohamed.abdalla@mail.utoronto.ca (MA)

**Data Availability Statement:** All relevant data are within the paper and its Supporting Information files. All software/code and tutorials are available at https://zenodo.org/record/6400074.

## Abstract

Genome wide association studies (GWASs) for complex traits have implicated thousands of genetic loci. Most GWAS-nominated variants lie in noncoding regions, complicating the systematic translation of these findings into functional understanding. Here, we leverage convolutional neural networks to assist in this challenge. Our computational framework, peaBrain, models the transcriptional machinery of a tissue as a two-stage process: first, predicting the mean tissue specific abundance of all genes and second, incorporating the transcriptomic consequences of genotype variation to predict individual abundance on a subject-by-subject basis. We demonstrate that peaBrain accounts for the majority (>50%) of variance observed in mean transcript abundance across most tissues and outperforms regularized linear models in predicting the consequences of individual genotype variation. We highlight the validity of the peaBrain model by calculating non-coding impact scores that correlate with nucleotide evolutionary constraint that are also predictive of disease-associated variation and allele-specific transcription factor binding. We further show how these tissue-specific peaBrain scores can be leveraged to pinpoint functional tissues underlying complex traits, outperforming methods that depend on colocalization of eQTL and GWAS signals. We subsequently: (a) derive continuous dense embeddings of genes for downstream applications; (b) highlight the utility of the model in predicting transcriptomic impact of small molecules and shRNA (on par with *in vitro* experimental replication of external test sets); (c) explore how peaBrain can be used to model difficult-to-study processes (such as neural induction); and (d) identify putatively functional eQTLs that are missed by high-throughput experimental approaches.

## Author summary

High-throughput assays are the cornerstone of modern drug discovery and a useful tool to translating the hundreds of genetic discoveries associated with human traits and disease

**Funding:** The author(s) received no specific funding for this work.

**Competing interests:** The authors have declared that no competing interests exist.

into functional understanding. All high-throughput assays can be described as empirical assessments of the activity of biological entities (e.g., genetic variation, DNA sequences, small molecules) by a standardized output, usually in the form of optically detectable labels (i.e., reporters), or more rarely, using (scalable) high-dimensional measurements (e.g., L1000, RNA-seq). Here, we introduce a modular and readily-extensible computational framework, called peaBrain, that leverages convolutional neural network architecture to enable *in silico* recapitulation of certain features of these high-throughput assays. We show that peaBrain can predict the expression of genes in a tissue-specific manner and outperforms regularized linear models in predicting the consequences of individual genotype variation. We further highlight the utility of the framework in predicting transcriptomic impact of small molecules and shRNA (on par with *in vitro* experimental replication of external test sets), explore how peaBrain can be used to model difficult-to-study processes (such as neural induction), and finally, identify putatively functional eQTLs that are missed by high-throughput experimental approaches.

## Introduction

Most reported disease-associated variation for complex traits lies in non-coding regions of the genome [1]. Despite advances in discovery and annotations of functional non-coding elements across the genome [2–5], characterising the consequences of non-coding variants remains a major challenge in human genetics. Prediction of the transcriptomic consequences of non-coding variation represents one solution [6–12], distinct from colocalization-based approaches that depend on the availability of genome-wide genetic associations [7,13–15], annotation-/frequency-driven prioritization of "functional" variants [16–19], and the use of non-human model genomes (so-called 'cross-species regulatory sequence prediction') [20]. Current methods of variant-expression prediction can be broadly divided into two classes: (a) methods that predict alterations in epigenetic and transcription factor binding sites (TFBS), such as DeepSEA [8], Bassenji [11], Enformer [21], and others [10,12]; and (b) methods that directly predict RNA abundance from genotype or sequence data, such as PrediXcan [6] and TWAS [9]. Methods in the former category poorly capture differences in transcript expression as a result of genotypic variation [8,10,11] and are relatively poor predictors of alterations in the histone code [8]; methods in the latter category are not able to identify which of the variants detected within an eQTL association locus are functional [6,9]. Recently, there has been development of a third class of models: *ab initio* sequence-based predictions using neural network architectures (e.g. ExPecto [22] and Xpecto [23]). While useful in predicting proximal mutations in promoters, ExPecto has a limited range for eQTL predictions (only 20kb upstream/downstream of transcription start sites) and the first step in the algorithm requires transformation of genomic sequences to epigenomic features (i.e. is not direct-from-sequence prediction and requires *a priori* annotation information). Furthermore, ExPecto was not designed with extensibility to other applications in mind (e.g. incorporating small molecule fingerprints or shRNA sequence to predict transcriptomic perturbations). DeepMind's Enformer side-steps the distance barrier by using transformers to incorporate longer sequences [21], but has yet to find utility in predicting differences between two individuals–likely because transformers lose positional information necessary for prediction. (Transformers enable modelling relationships between distant regulatory motifs in a genomic sequence, but does not retain information about their respective positions.) The final example in this class is Xpecto [23], which uses much of the code of an earlier version of peaBrain released with the pre-print (and in fact cites

the pre-print associated with this manuscript), but does not incorporate the modular design/ extensibility to other applications, is also limited by its ability to capture relevant information from distant regulatory sequences, and cannot predict differences in transcription between two individuals.

To address these concerns, here, we introduce a single framework, called **p**romoter-and-**e**nhancer-derived **a**bundance (**pea**Brain) model, which consolidates these approaches. Within the peaBrain framework, the transcriptional machinery of a tissue is modelled computationally as a two-stage process. Stage 1 is a single model in which peaBrain predicts the mean abundance of each gene in a given tissue from DNA sequences, optionally annotated with epigenetic and genomic annotations. Stage 2 incorporates the transcriptomic consequences of genotype variation to predict individual abundance of any given gene; that is, it generates a gene- and tissue-specific model sensitive to individual variation. Stage 2, unlike existing neural network models, does not depend on the availability of epigenetic and genomic annotations for training or prediction (i.e. is purely sequence based) and can incorporate 1Mb window around the TSS for any gene (50x larger than ExPecto [22] and similar models). Either stage is readily extended for closely-related applications (such as predicting transcriptomic impact of small molecules and shRNA).

We demonstrate that the convolutional neural networks (CNNs) underlying this framework can capture the majority of variance (cross-validated cv-r$^2$ >50%) in the mean abundance of genes across most GTEx tissues (Stage 1), with utility in a diverse set of tasks (such as identifying somatic mutations with high-impact consequences or pinpointing the functional tissues underlying GWAS signal from complex traits). In **S1**–**S3** **Text**, we highlight a variety of case (proof-of-concept) applications of the Stage 1 peaBrain model, including but not limited to: investigating the role of DNA and histone modifications in difficult-to-study processes (such as neural induction) and incorporating small molecule fingerprints (or shRNA sequences) to predict the transcriptomic impact of small molecules (or shRNA molecules). We further show that CNNs–using sequence alone and no genomic/epigenetic annotations–outperform linear models and other neural network architectures in predicting the consequences of genotype variation (Stage 2). In EBV-transformed lymphocytes (LCLs), we demonstrate that the estimated peaBrain variant effects correlate more strongly with coefficients from the univariate eQTL analysis, compared to log-skew effect estimates obtained from massively parallel reporter assays (MPRAs) [24] and bi-allelic targeted STARR-seq (BiT-STARR-seq) [25], or log fold changes (logFC) of perturbed epigenetic states from DeepSEA [8]. To highlight the utility of the Stage 2 models, we identified putatively functional eQTLs in LCLs that are missed by experimental high-throughput approaches that characterise variant function, such as MPRAs, BiT-STARR-seq, and high-definition reporter assays (HiDRA) [26].

## Results

### peaBrain captures >50% of the variance in mean gene abundance

To predict the tissue-specific mean abundance of genes (Stage 1), we leveraged the reference genome [27]. For each gene, as input, we generated a 1-dimensional (1D) matrix centred on the region around the annotated transcription start site (TSS). By varying the length of the input sequence, the 4kbps promoter (2kbps upstream and 2kbps downstream of the annotated TSS) was determined as the best-performing length for predicting the tissue-specific mean gene abundance in the GTEx dataset, outperforming 2kbps and 6kbps promoter sequences (see **Methods** and **S1 Fig**). We used one-hot encoding (four channels) to represent the four DNA letters (A, T, C, G) in the reference genome (4 channels) (see **Methods**). The model

output was the corresponding predicted mean RNA abundance of that gene, after rank-transformation to normality.

We applied this framework to all tissues from the GTEx dataset [28], constructing three classes of models: (a) using DNA sequence alone (class-A); (b) using DNA plus epigenomic annotations not specific to any tissue or cell type (i.e. non-specific annotations) (class-B); and (c) using DNA combined with both non-specific and tissue-specific annotations (class-C). For class-B models (DNA + non-specific epigenomic annotations), we incorporated 28 channels of binary sequences that represent epigenomic (and related) annotations that are not specific to any cell type or tissue (curated by the authors of LD Score Regression [29]; see **Methods** for details). For class-C models (DNA + tissue-specific epigenomic annotations), we added additional channels corresponding, for those tissues where such data were available, to the consolidated epigenomes from the Epigenomics Roadmap, including tissue-specific peaks from H3K4me1, H3K4me3, H3K9ac, H3K9me3, H3K27me3, and H3K36me3 ChIP-seq experiments, and experimentally-derived DNase hotspots [30].

We observed that DNA-only (class-A) models captured nearly a fifth of the variance in mean gene abundance across all GTEx tissues (10-fold cross-validated median cross-validated-$r^2$ [cv-$r^2$] values across all tissues = 17%). Addition of non-specific regulatory annotations (class-B models) markedly improved model performance across all tissues (median cross-validated cv-$r^2$ = 45%; **Fig 1**). (We average the cv-$r^2$ across all 10-folds within a tissue and use the median across all tissues to assess global performance; see **Methods**.) For example, for EBV-transformed lymphocytes, the 10-fold cross-validated average cv-$r^2$ is 56% for the class-B model (DNA + non-specific annotations) compared to the 15% in the corresponding class-A model (DNA-only). Addition of tissue-specific annotations further improved model performance, such that class-C models (DNA + tissue-specific annotations) captured more than half the variance for almost all GTEx tissues where such data were available (**Fig 1**).

These results are suggestive that differences in mean abundance between genes are largely encoded in differences between core promoter elements and interacting regulatory factors encoded in the model weights, rather than a consequence of non-transcriptional downstream regulation (e.g. silencing by small non-coding RNAs). This is broadly consistent with anecdotal experimental evidence [31]. Explicitly incorporating experimental transcription factor binding site (TFBS) annotations has limited effect on performance (median cross-validated $r^2$ = 23%), when compared to the complete class B model with epigenetic/histone marks and chromatin annotations (median cross-validated $r^2$ = 46%; **S1 Text**). This suggests explicitly encoding TFBS annotations is largely redundant and that epigenetic and genomic annotations add information to that contained in the DNA sequence to substantially improve predictive performance. Importantly, this performance was only accomplished using the convolutional neural network architecture of peaBrain; experimental models that we generated using regularized linear models (as a simple baseline model) fitted with stochastic gradient descent exhibited poor performance. In fact, for these linear models the 10-fold average cv-$r^2$ was negative, indicating that the cross-validated predictions of the model fitted on the training data are worse than predicting the mean of the test set (see **Methods** and **S1 Text**). We also describe comparisons of the peaBrain CNN approach with other methods in **S1 Text**.

## Testing on expression data from microarray platforms and using orthologous promoters in non-human Hominid primates further validates performance and generalizability of the peaBrain Stage 1 models

One useful metric in assessing generalizability of any model is its ability to predict on gene expression measured by other technology platforms (i.e. other than RNA-seq in this case).

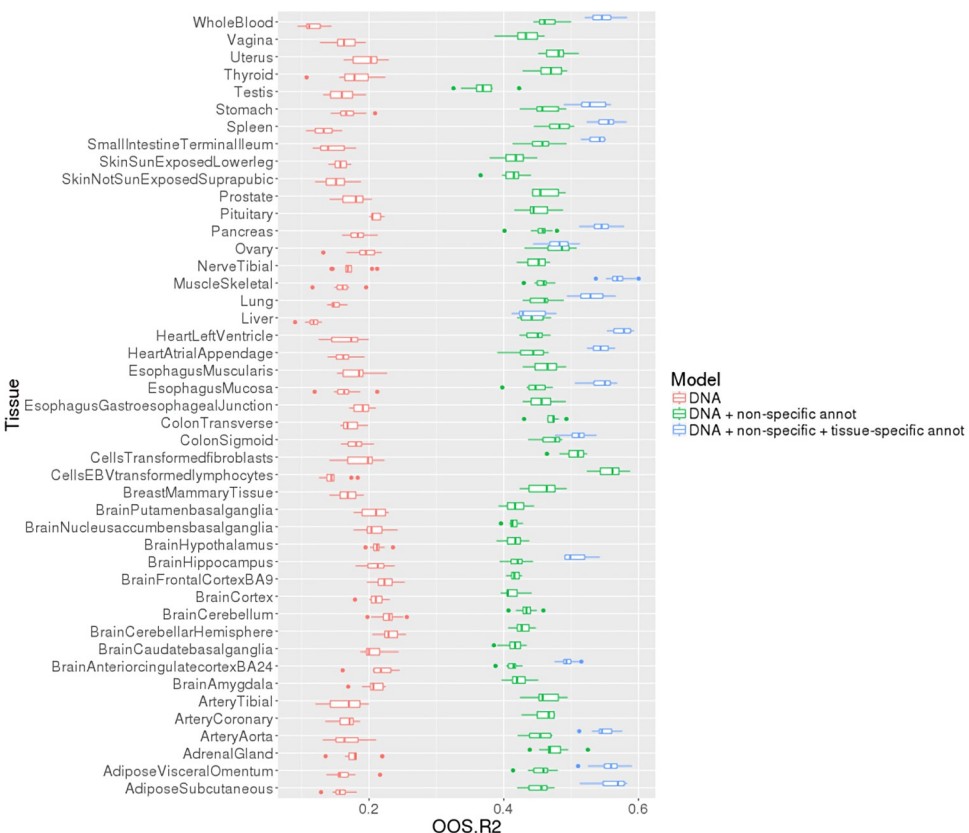

**Fig 1. Incorporating genomic and epigenetic annotations improves the performance of peaBrain to predict the normalized mean abundance across all GTEx tissues.** The 4kbps promoter sequence, when annotated with tissue specific annotations, is sufficient to predict the majority of variance in mean expression in most tissues, ordered alphabetically from the x-axis. The boxplots highlight the distribution of the 10-folds used to cross-validate model performance. Prediction using regularized linear models performs considerably worse (10-fold cross-validated $r^2 < 0$; **S1 Text**). **Abbreviations:** OSS.R2, out-of-sample $r^2$.

This is a useful benchmark that indicates the model is not inherently biased towards the underlying technology from which the training data was derived. We subsequently sought to assess the performance of the peaBrain DNA-promoter reporter assay on external datasets in three human tissues (expression measured by microarray; Spearman's rho between predicted and external platform = 51%-63%; **Fig 2**) and on its ability to predict expression of orthologous promoter sequences in four non-human Hominid primates in two different tissues (rho = 5%-22%). For the non-human Hominid primates, we used the promoter sequences from their respective genomes and the human-trained peaBrain models. In other words, the human-trained peaBrain models were able to capture a large portion of the variance in expression, which suggests that there is conservation of transcriptional cascades across the evolutionary clade. Spearman's rho was used because peaBrain was trained using RNA-seq and the test data sets were generated on microarray platforms and/or different species. This means that the magnitude of expression for any given gene is different; however, the ranks of the genes would be expected to be largely consistent. All together, these results are suggestive that differences in mean abundance between genes are largely encoded in differences between core promoter elements and interacting regulatory factors encoded in the model weights. With analyses performed in **S1 Text** and briefly noted above, these experiments suggest that expression-based HTAs (e.g. small molecule screens) can be recapitulated by extending this DNA reporter assay;

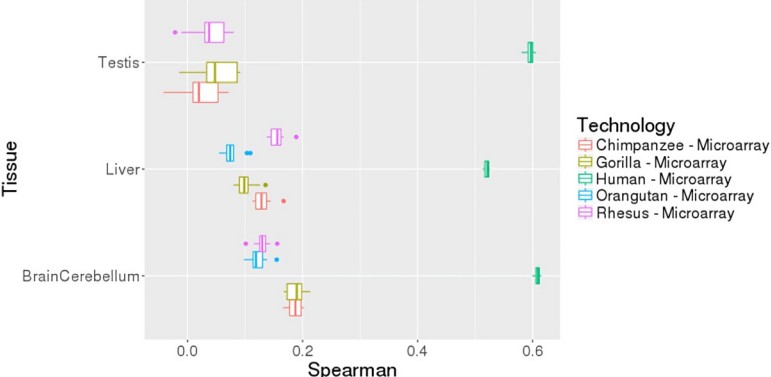

**Fig 2. Boxplots of the cross-validated peaBrain models when trained on human GTEx RNA-seq data and tested on microarray expression data from the Hominid lineage (including humans).** The performance on non-human hominid species suggests that the peaBrain model–when trained exclusively on human–can also shed insight the transcriptional machinery of related species (both extant and extinct). This could be useful, for instance, when tracing the evolution of regulatory cascades in the Hominid lineage.

that is, the information regarding the transcriptional machinery of any given tissue or cell type is encoded in and can be extracted from the promoter sequence (and the corresponding gene expression).

## peaBrain score outperforms existing measures in predicting disease-associated variants and in predicting allele-specific transcription factor binding

Having demonstrated the predictive ability of the model (Stage 1), we were interested in using peaBrain to generate a non-coding impact metric, which captured the impact of each position in the core promoter sequence on the expression of each gene. We defined the impact of each position as the absolute difference in abundance between the original promoter sequence and a modified promoter sequence where all the information for that site (including epigenetic and genomic annotations) is set to zero (**Fig 3**). To facilitate comparison across tissues, we performed this analysis using the class-B models, since the non-specific epigenetic and genomic annotations were, by definition, available for all tissues. Across all GTEx tissues, the non-coding impact metric correlated with variant-specific conservation scores derived from multiple alignments of 99 vertebrate genomes to the human genome [27] and represented by phylogenetic p-values (phyloP) (see **Methods**). Briefly, these phyloP nucleotide conservation scores are based on an alignment and a model of neutral evolution [27]: a more positive value indicates conservation or slower evolution than expected, with the magnitude of the phyloP score corresponding to the -log p-values under the null hypothesis (i.e. neutral evolution). For every unit of absolute magnitude increase in impact, we observed an average increase of 8.95 in phyloP scores, indicating increased conservation (8.95 order-of-magnitude difference in the–log10 p-value; **S1 Table**). Equivalently, for every unit increase in phyloP, we observed an approximately 0.1 absolute magnitude change in the average normalized expression of the affected gene (i.e. peaBrain impact score); again indicating that if a site is more conserved, it has a larger impact on expression. While this positive trend between conservation and impact on expression was consistent across most GTEx tissues, there were exceptions: in the nucleus accumbens (basal ganglia), noncoding transcriptomic impact was correlated with accelerated evolution (**S1 Table**). These results were consistent, albeit weaker, after rank-normalization of both the phyloP and peaBrain scores (**S1 Table**).

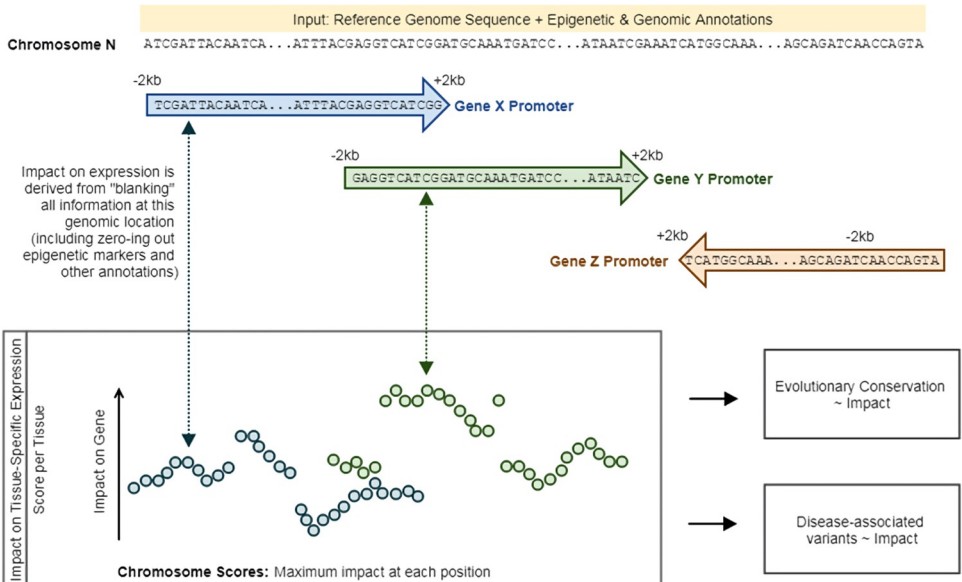

**Fig 3. Schematic illustration of how the non-coding impact score was calculated.**

This overall positive correlation between peaBrain impact and phyloP represents a direct equivalence between evolutionary conservation and impact on gene abundance. Most well-established non-coding impact measures (e.g. CADD [16] and Eigen [17]) indirectly capture transcriptomic consequences by modelling evolutionary conservation measures, allele frequency, and/or functional non-coding consequence annotations. However, the peaBrain-derived impact metric directly assesses the contribution of a genomic position on mean expression. Importantly, since the metric is independent of curated consequence and disease annotation databases–as it is trained solely on expression from "healthy" tissues–it provides an unbiased estimate of the information content and deleterious impact of variation at any genomic position in the core 4kbps promoter sequence.

Having established the correlation between peaBrain impact and evolutionary constraint, we were interested in assessing the utility of peaBrain-derived scores to interrogate disease-associated variants. We compared the performance of the non-tissue-specific peaBrain score (see **Methods**) to two other non-coding metrics (CADD [16] and Eigen [17]) across a series of tasks (**tasks A-C**; all tasks are summarized in **S2 Table**).

First, we made use of data on disease-related variation from the Catalogue of Somatic Mutations in Cancer [COSMIC] [32], limited to the census gene set which defines a set of genes with somatic mutations causally implicated in human cancer (see **Methods**). In **task A,** we assessed the predictive capacity of the non-coding metric to identify positions with non-zero incidence of cancer-associated somatic mutation (n = 5268), among all genomic positions within the 4kbps core promoter sequences of COSMIC census genes (approximately 2.15 million positions), using a simple logistic model. The logistic coefficients give the change in the log odds of the outcome (i.e., presence or absence of somatic mutation) for a one-unit increase in the non-coding score. In **task B,** we similarly assessed the predictive capacity of the non-coding metric to identify, using the same COSMIC data set, positions with recurrent cancer-associated somatic mutations (n = 544) when contrasted to positions with non-recurrent cancer-associated somatic mutations (n = 4724). The focus on cancer-associated somatic mutations allowed us to circumvent linkage disequilibrium (LD) confounding. Patterns of recurrent non-coding somatic mutations, across all tumours in these genes, provide a coarse

indicator of the functional transcriptomic impact of non-coding genomic positions. Both tasks were modelled with the allele frequency and phyloP conservation incorporated as covariates (see **Methods**). We subsequently assessed the significance of the logistic model coefficients for each of the non-coding metrics across the two tasks (**Table 1**). Only the non-specific-peaBrain score, derived from scores across all GTEx tissues (average across all tissues and positions), was positively and significantly predictive for both tasks (**Table 1**). Significance was assessed using the default two-tailed p-value corresponding to the z ratio based on the Normal reference distribution (**Table 1**; see **Methods**). The non-specific peaBrain-derived metric was useful in isolating genomic positions with non-zero incidence of somatic mutations across all positions in the promoters of COSMIC consensus genes (**task A;** coefficient point estimate = 29.36; 95% confidence interval [ci] (16.63, 41.97)), and could further delimit positions with recurrent somatic mutations (**task B**; coefficient = 102.96 [64.58, 140.72]). Eigen was significantly predictive for **task A** (coefficient = 0.10 [0.08, 0.12]), but not for **task B** (0.08 [-0.01, 0.17]). CADD exhibited the opposite trend between **tasks A and B**: negatively predictive of genomic positions with non-zero incidence of somatic mutations (coefficient = -0.05 [-0.08, -0.02]), but positively predictive of positions with recurrent somatic mutations (coefficient = 0.17 [0.06, 0.28]; **Table 1**). Thus, the non-coding peaBrain-derived metric appears to better characterize the pathogenicity and putative functionality of non-coding variants with transcriptomic consequences in the core-promoter sequences, providing additional information to that found in allele frequency or evolutionary constraint metrics and with performance better than other established non-coding impact scores.

**DNA sequence, annotated with experimentally-derived TFBS, from core promoter sequences are insufficient to predict mean abundance with high accuracy–epigenetic/histone markers contain the bulk of the information and are not readily accessible from the DNA sequence alone.** We were interested in determining the contribution of epigenetic/histone makers, alongside more general genomic annotations (such as coding sequences), in predicting the mean abundance of genes. In particular, we wanted to explore whether the DNA

**Table 1. Tabulated statistics (to two decimal places) from the logistic models for the three non-coding metrics from tasks A-C.**

| Metric | Task | Logistic Coefficient | L Bound | U Bound | p-value |
|---|---|---|---|---|---|
| peaBrain | **A** | **29.36** | **16.63** | **41.97** | **5.56 x10$^{-6}$** |
| | **B** | **104.50** | **66.05** | **142.31** | **7.66 x10$^{-8}$** |
| | **C** | **35.39** | **12.00** | **58.67** | **2.95 x10$^{-3}$** |
| CADD | **A** | **-0.05** | **-0.08** | **-0.02** | **1.57 x10$^{-3}$** |
| | **B** | **0.17** | **0.06** | **0.28** | **2.83 x10$^{-3}$** |
| | C | 0.06 | -0.03 | 0.16 | 0.20 |
| Eigen | **A** | **0.10** | **0.08** | **0.12** | **< 2 x10$^{-16}$** |
| | B | 0.06 | -0.003 | 0.12 | 6.65 x10$^{-2}$ |
| | C | 0.04 | -0.002 | 0.08 | 0.07 |

**Task A** assesses the predictive capacity of the non-coding metric to identify positions with non-zero incidence of cancer-associated somatic mutations in the core promoter regions. **Task B** assesses the predictive capacity of the non-coding metric to identify positions with recurrent cancer-associated somatic mutations, among all positions with at least one somatic mutation. **Task C** assesses the predictive capacity of the non-coding metric to identify variants within the 4kbps core promoter with allele-specific binding (for a subset of positions for which data was available). All three tasks were assessed using simple logistic models, with the allele frequency and phyloP incorporated as covariates. Positions without a phyloP score were excluded from model fitting (see **Methods**). peaBrain is the only non-coding metric with significant coefficients for all three tasks; we used a non-tissue-specific peaBrain score to facilitate comparison with the tissue-agnostic CADD and Eigen scores (see **Main Text**). The bounds for the 95% confidence interval, obtained by profiling the likelihood function, are tabulated, with significant coefficients denoted in bold. peaBrain's impact score has the same "units" as normalized expression. Eigen and CADD are on arbitrarily-normalized scales (hence the difference in coefficient magnitudes); normalizing Eigen and CADD resulted in loss of significance for most tasks. **Abbreviations:** L, lower; U, upper.

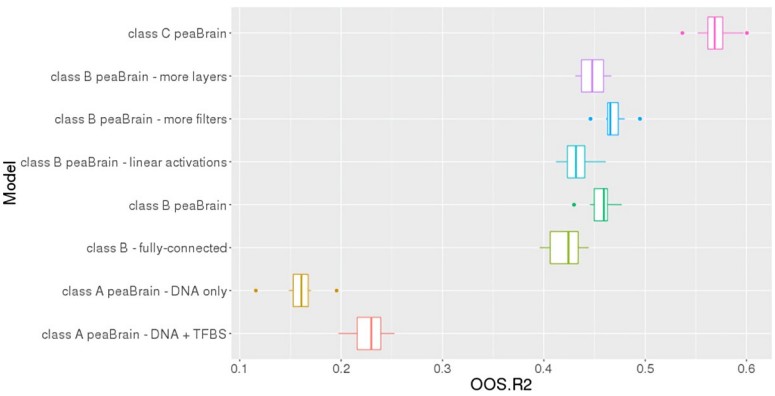

**Fig 4. Boxplots of 10-fold cross-validated cv-r$^2$, as assessed in skeletal muscle.** Performance as assessed for class A models (labelled as "class A peaBrain–DNA only"), class A with TFBS annotations (labelled as "class A peaBrain–DNA +TFBS"), class B models with tissue-agnostic annotations ("class B peaBrain–CNNs"), fully connected neural networks ("class B–fully-connected"), class B models with linear activation functions ("class B peaBrain–linear activations"), class B models with increased number of layers ("class B peaBrain–more layers"), class B models with increased number of filters ("class B peaBrain–more filters"), and class C models with tissue-specific annotations ("class C peaBrain").

sequence alone was sufficient to predict expression. We noted that increasing the number of convolutional layers or the number of filters did not improve model performance (**Fig 4**). Explicitly incorporating TFBS into the model (i.e. annotating the DNA only and explicitly with TFBS) only improved performance slightly (cv-r$^2$ = 23%), and was still considerably worse than the full class B model with epigenetic/histone marker annotations (cv-r$^2$ = 46%; **Fig 4**). (Class-A DNA-only models had an average cv-r$^2$ of 16% for skeletal muscle; class-C models annotated with tissue-specific information had an average cv-r$^2$ of 57%.) The TFBS were collected from the Gene Transcription Regulation Database (GTRD) v17.4 with data on 476 human transcription factors and included peak calling with four different software (MACS, SISSRs, GEM, and PICS). In addition to including the processed peak calls, we also incorporated clusters (i.e. peaks merged for the same transcription factor but under different experimental conditions) and meta-clusters (i.e. non-redundant peaks synthesized from all four methods). This absence of improvement suggests that peaBrain model already recognizes many of the TFBS; identified by the convolutional filters inherent to the model architecture. These results indicate that experimentally-derived epigenetic and genomic annotations add information to that contained in the DNA sequence alone. This is broadly consistent with the observation that other convolutional neural networks models like DeepSEA are better at predicting TFBS (median AUC = 0.958) than at predicting histone modifications (median AUC = 0.856) [8].

**peaBrain score out-performs existing measures in predicting allele-specific transcription factor binding.** As with **tasks A and B,** we compared the performance of the non-tissue-specific peaBrain score to predictions by CADD and EIGEN in predicting allele-specific binding, after accounting for allele frequency and evolutionary conservation. We assessed performance of the three non-coding metrics across 6675 sites in core promoter regions after filtering for duplicate sites [33]; 1896 of which exhibited allele-specific binding at an unadjusted binomial p < 0.05 (see **Methods**). We noted that only peaBrain impact score was significantly predictive of allele-specific binding sites (coefficient = 35.38 [12.00, 58.67]; p = 0.003; see **Table 1**); relaxing the binomial p-value threshold (i.e., increasing the number of sites considered as allele-specific) brings the other non-coding metrics to significance. peaBrain's discriminative ability to identify allele-specific binding sites is consistent with our earlier observation

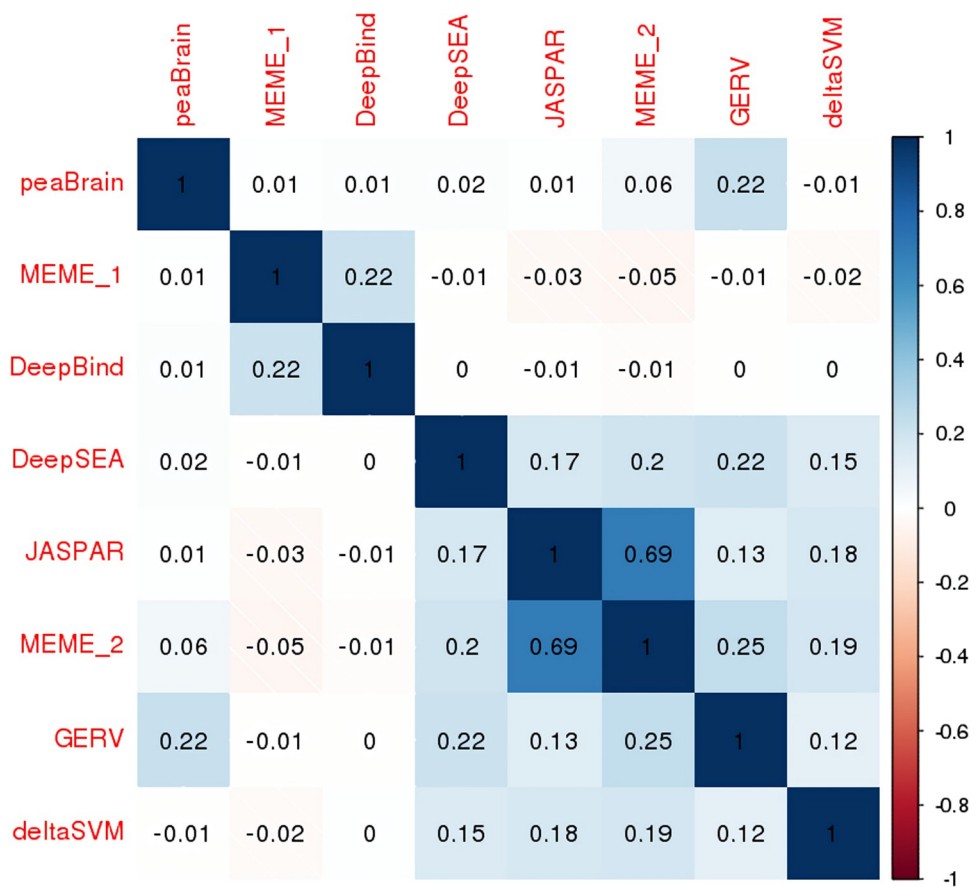

**Fig 5. Rank correlation plot for TF-binding algorithms and the peaBrain impact score.** JASPAR, MEME_1 and MEME_2 are PWM-approaches.

that explicitly adding TFBS annotations did not improve the model. Notably, peaBrain's ability indicates that average expression of all genes in a single tissue and the reference genome is sufficient to learn both TFBS and allele-specific binding.

To further investigate peaBrain's ability to identify allele-specific binding sites, we compared peaBrain impact scores to predictions by methods specifically designed to predict TFBS, including two neural-network methods (DeepBind [34] and DeepSEA [8]), two kmer-based variant scoring methods (gkmSVM [35] and GERV [36]), and three position-weighted matrices (PWM)-related methods [33]. These methods depend on modelling TF ChIP-seq data in various ways and may have multiple models for the same TF. After confirming the predictive ability of these methods to identify allele-specific binding sites, we noted that peaBrain scores positively correlated only with GERV measures, a kmer-based variant scoring algorithm (**Fig 5**). Unlike the other methods, peaBrain (and GERV) do not assume the existence of canonical motifs and learn TFBS by modelling sequences (or kmers) directly (i.e. not simply by modelling the absence or presence of a ChIP-seq peak). In contrast, for both DeepBind and DeepSEA, we noted positive correlation with at least one PWM-method. These methods generally assume the existence of canonical TF binding sites and predictions are based on the extent of perturbation of those motifs. While this comparison is limited to variants for which data was available, the peaBrain results suggest that explicitly characterizing TF motifs is not necessary to understand the consequences of sequence variation on TF binding and transcriptional dysregulation.

## Tissue-specific peaBrain scores can identify the functional tissues underlying GWAS signals from complex traits

For **tasks A-C**, we have used the non-tissue-specific peaBrain score (average of score, per position, across all tissues) to facilitate comparison with the other tissue-agnostic impact metrics. However, we sought to investigate advantages of tissue-specific impact scores. In particular, we wanted to highlight how tissue-specific scores could allow us to identify functional tissues associated with GWAS signal from complex traits (**task D**). We hypothesized that the "true" functional gene(s) downstream of a GWAS locus ("hit") would have, on average, higher peaBrain impact scores for the tissue in which the gene is likely to act, given that >50% of the variance in mean gene abundance can be explained by the promoter sequence. In other words, we hypothesized that genes associated with a given phenotype (e.g. total cholesterol) are also likely to be transcriptionally perturbed in the underlying functional tissue (e.g. liver), which we can detect with tissue-specific peaBrain scores.

For brevity, we selected 4 quantitative traits [37] (total cholesterol, LDL, HDL, and triglycerides) for which the (primary) putatively causal tissue is well-established and included in the GTEx dataset. Using HESS [38], we calculated the local SNP-heritability from the relevant GWAS summary statistics, while accounting for linkage disequilibrium. For European populations, HESS partitions the genome into 1703 approximately-independent LD blocks (average length = 1.6Mb) [38]. For each block (or "locus"), we calculated the tissue-specific peaBrain impact score for each GTEx tissue; the locus peaBrain score is defined as the average of the tissue-specific peaBrain scores at all positions (with a score) within that locus. We subsequently performed a regression of the rank-transformed local SNP-heritabilities as a function of the rank-transformed peaBrain locus scores to minimize bias caused by outlying loci and assessed significance for the linear model coefficient (n = 45 tests for each GTEx tissue per phenotype; see **Methods**). As a baseline benchmark, we compared our results to tissue predictions made using the tissue trait concordance (RTC) score [39], which was adapted to calculate the probability that a GWAS-associated variant and an eQTL are co-localized and weighted by the extent of tissue sharing for the given eQTL to obtain tissue-causality profiles for each trait. Across all tested traits, we noted the peaBrain framework was better at identifying putatively causal tissues than simply using the RTC-/eQTL-based method (**S3 and S4** Tables). For LDL, using the peaBrain framework, the top five tissues (ranked by nominal p-value) were: EBV-transformed lymphocytes, visceral adipose, fibroblasts, liver, and terminal ileum (small intestine); all were significant after Bonferroni adjustment with p-values tabulated in **S3 Table**. In contrast, with the RTC-based method, the top five tissues were: sun-exposed skin (from lower leg), pancreas, fibroblasts, tibial nerve, and cerebellar hemisphere (brain). This was consistent across all tested traits (e.g. for HDL, liver ranked 3rd using the peaBrain framework and 32nd using the RTC-based method; **S3 and S4** Tables). The superior peaBrain performance suggests inherent limitations to eQTL-based methods that are sidestepped by the Stage 1 peaBrain framework, which depends only on the average expression of all genes in a single tissue and the reference genome. Notably, peaBrain is independent from the number of eQTLs identified per tissue and the number of genome-wide significant hits for a given trait, which are both limitations for eQTL-GWAS co-localization methods (such as the RTC-based framework). Further confirming this hypothesis, we observe that GenoSkyline-Plus [15], a tool that uses expanded set of epigenomic and transcriptomic annotations to produce high-resolution, single tissue annotations (rather than eQTLs alone), identifies that all 4 lipid traits shared a similar enrichment pattern in liver, adipose, and monocyte tissue (**S4 Table**), consistent with the causal relationship among these traits observed with the peaBrain analysis. Interestingly, both peaBrain and GenoSkyline identified liver, monocytes, adipose, and gastrointestinal tissue as

the primary tissues for total cholesterol, in the same rank order (S4 Table)–further supporting the validity of this approach.

Having validated the peaBrain Stage 1 approach and its utility in a diverse set of tasks, in S1 Text, we highlight how activations of the penultimate layer of the peaBrain model can be used as a continuous and compressed representation (i.e. embedding) of genes. These embeddings, or equivalently, neural activations, capture both the annotated DNA (input) and its additive contributions to tissue-specific abundance (output) in a compressed form amenable to downstream analyses (such as network-based analyses). These embeddings display interesting properties (see S1 Text), including the encoding of correlation information and membership to pathways/curated gene sets. Importantly, these embeddings are in a linear space, such that the pairwise cosine similarity between these dense gene representations is proportional to the measured RNA-seq correlation between the gene pair. In other words, co-regulation and co-expression may be discovered by leveraging linear structure within the embeddings (e.g. adding embeddings of two genes to discover their co-expression with a third).

## Stage 1 peaBrain model is designed with extensibility in mind and can be applied to a diverse set of applications

To further highlight the utility and extensibility of the Stage 1 peaBrain models, we provide two case (proof-of-concept) studies.

In S2 Text, we demonstrate the utility of this peaBrain DNA reporter model in investigating the role of DNA and histone modifications in difficult-to-study processes, such as neural induction. By modelling the transitions between consecutive neural progenitor cell stages, we identified the subset of genes in each stage-specific differentiation whose expression is directly altered by epigenetic modifications in their promoter sequences. Among the genes identified in the differentiation of neuro-epithelial and mid-radial glial cells, we note significant enrichment of genes implicated in schizophrenia, autism, bipolar, and depression (1.5–2.6 fold enrichment; Fisher's exact $p < 0.05$); a trend that becomes more pronounced when limited to genes implicated using genetic associations alone. With nearly 5–10% labelled as transcription factors (1.53–3.24 fold enrichment), this cross-disease enrichment provides a putatively causal mechanism early in neural development for the shared genetic correlation between these psychiatric phenotypes–an observation that is difficult to extract from GWAS data alone and is missed by differential peak/gene expression analyses.

In S3 Text, we show, with a small extension to this *in silico* DNA reporter model to incorporate small molecule fingerprints, this network architecture can also be used to predict the transcriptomic impact of small molecules in cell lines, with as few as 600 molecules (24 hours post-exposure). Predictive performance of this small molecule screen is on par with *in vitro* experimental replication of external test sets, allowing us to impute expression for all clinically approved molecules in the ChEMBL database. By training on the imputed expression profiles for all molecules first approved prior to the year 2000, we are able to retrospectively identify molecules that would later be assigned to the corresponding indications (post-2000), with 64–86% increase in F1-scores (i.e., the harmonic mean of precision and recall, a measure of model performance and accuracy that is robust to class imbalance), 22–63% increase in precision, 75–94% increase in recall, and 13–19% increase in area under precision-recall curve (AUPRC), compared to current chemoinformatic/molecule structure-based approaches.

Using a nearly identical model and with better performance, we can also predict the transcriptomic consequences of shRNA olgio-sequences (i.e. gene knockdowns). We use this shRNA peaBrain model to predict the consequences of 330,617 unique shRNA oligo-sequences (targeting 19,992 human genes) to enable the inference of regulatory networks, at a

precision of 66–77% (AUROC = 87–93%, AUPRC = 39–52%, F1-score = 14–45%). We leverage these models to identify 212 new transcription factor-target interactions, of which 83% are supported by experimental ChIP-seq evidence.

## peaBrain model architecture can be leveraged to predict the transcriptomic consequences of individual variation

Having shown the utility of the Stage 1 peaBrain model, we extended the peaBrain model to incorporate the transcriptomic consequences of individual genotype variation (**Stage 2**). Given whole genome sequencing data of a group of individuals (such as GTEx participants), we sought to assess the ability of this extended peaBrain model to predict the tissue-specific expression profile of each individual, and to identify putatively functional variants within the sequence (**Fig 6**).

To do this, we constructed, for each gene and in each tissue, an extended peaBrain model that takes individual genome sequence as input and predicts the tissue-specific expression of the corresponding gene as output. (For stage 2 analyses, we did not make use of individual level epigenomic and regulatory annotations as these were not available.) More concretely, unlike stage 1 models, for a single gene, stage 2 models predict the difference between the expression of two individuals as a function of the difference in the sequences between the two individuals (for the given gene; see **Methods**). By jointly modelling the input "difference" sequence in a non-linear manner (as a result of the activation function of the CNN), we hypothesized that we would capture information relevant to *cis*-heritability missed by linear models (such as distance to TSS sites and the pairwise relationships between variants), and be able to prioritize functional variants with transcriptomic consequences solely from the DNA sequence. This additional information is modelled by using the "difference" sequence as input, rather than the dosage in variation. (Stage 2 peaBrain models were trained separately from Stage 1 models, but share similar architectures; see **Methods**.)

Consistent with evidence from eQTL studies [40], we noted the 4kbps core promoter used in Stage 1 did not capture enough cis-heritability as estimated by constrained GCTA [41] and thus was not sufficiently informative for this predictive task. In LCLs, for example, using the

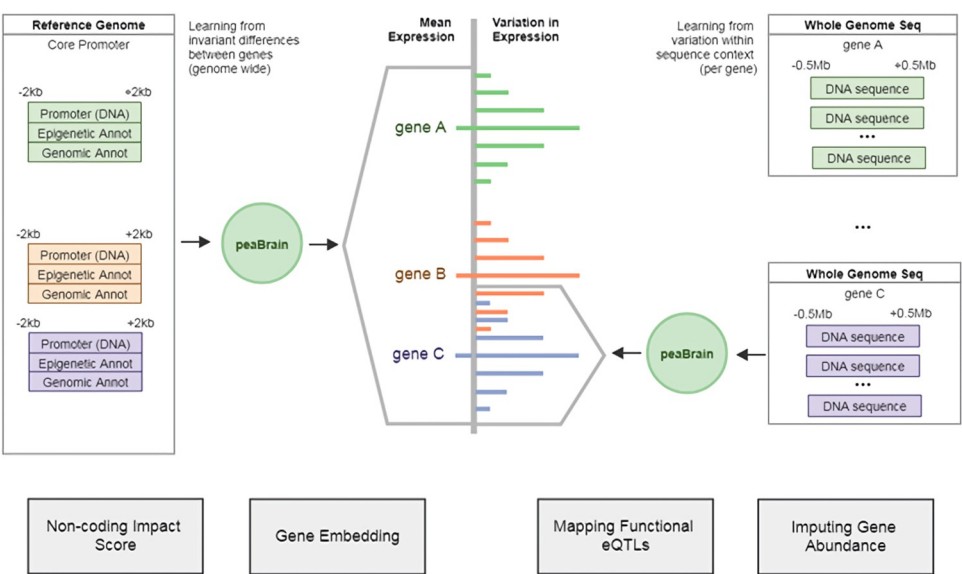

**Fig 6. Schematic illustration comparing Stage 1 and Stage 2 peaBrain models.**

4kbps core promoter, genes with significant non-zero heritability (p < 0.01; n = 1066) had a median heritability of 0.136. We selected a 1Mbps input length, centred on the annotated TSS (0.5Mbps upstream and 0.5Mbps downstream), as a compromise between computational tractability (extending the sequence entails more computational expense) and biological relevance (the potential to capture additional narrow-sense heritability with extended intervals). Using the 1Mbps input sequence, genes with significant non-zero heritability (p < 0.01; n = 816) had a median heritability of 0.270; nearly twice the heritability captured with the core promoter 4kbps sequence. Importantly, our symmetric 1Mbps window likely contained >95% of cis-eQTLs; in the GTEx dataset, the 95th percentile for absolute distance of cis-eQTLs from their target transcript TSS was 441,698bps [28]. Complete analysis of a 1Mbps interval (including 5 different train/test splits) for a single gene in a single tissue and 94 individuals, if run sequentially on a CPU, required 15 days with 14 GB of memory. Limited to the genes with significant non-zero heritability in LCLs (n = 816), on 600 cores, the complete analysis took approximately a month. (Stage 1 peaBrain models only required several hours.) Prior to training, for each individual, we re-constructed the 1Mbps input sequence from the variants called from whole genome sequence (WGS) data (see **Methods**). Exploring the peaBrain architecture, fine-tuning the model parameters, and deploying the models was conducted on NVidia's P100 GPUs (see **Methods** for details); the bulk of the training, however, was run on CPUs.

To assess peaBrain's performance in predicting individual variation in RNA expression levels in comparison to other widely-used *in silico* methods and experimental assays (elastic net [6], DeepSEA [8], MPRA [24], BiT-STARR-seq [25], and HiDRA [26]), we designed four tasks (**tasks E-H**; described below and summarized in **S2 Table**). For the comparison with elastic net, in line with other recent studies in the field [9], we restricted performance analyses to a set of genes with significant non-zero narrow-sense *cis*-heritability (henceforth, simply referred to as heritability) in LCLs as estimated by constrained GCTA [41] (limited to the 1Mbps input sequence; p<0.01; see **Methods**). By limiting analysis to genes with detectable *cis*-heritability, we can make more meaningful conclusions about the comparative performance of the different methodologies. We restricted analysis to LCLs to enable comparisons with empirical data (**tasks F-H**) and to reduce the compute burden. (We do not perform transcriptome-wide case-control analyses due to the compute resources required as highlighted above.)

## peaBrain identifies functional architecture that is inaccessible with current high-throughput experimental assays

First (**task E**), we compared the predictive performance of peaBrain to that of a regularized linear model (an implementation of elastic net identical to that used in PrediXcan). RNA-seq samples from the GTEx dataset (n = 94 individuals after filtering) were pre-processed, residualised to account for cryptic relatedness, biological confounders, and technical variance, and rank transformed to normality (see **Methods**) before modelling. Model performance for both linear models and peaBrain was assessed by generating cv-$r^2$ for 5% of individuals randomly withheld from training and unrelated to individuals in the training set (repeated 10 times; see **Methods**). For each of the 816 genes with non-zero heritability (GCTA p < 0.01), we calculated the 95% confidence interval for the cv-$r^2$, defining a gene as successfully predicted if the entire cv-$r^2$ confidence interval exceeded zero to ensure we only consider genes with high-confidence models. Whilst regularized linear models were able to capture cis-heritability for 28 of the 816 genes, the equivalent number for peaBrain was 113 (explanation for difference in performance is discussed below). *Cis*-heritability for 3 genes was captured by both models, with the cv-$r^2$ confidence interval largely overlapping (**S5 Table**). **S5 Table** also tabulates the performance metrics (confidence and point estimates for cv-$r^2$ from both classes of models) and

estimated GCTA heritability for all genes. This preliminary analysis highlights one key difference between linear models (e.g., elastic net) and peaBrain–peaBrain trades increased computational cost for better predictive performance on smaller to medium-sized datasets. As we will discuss further, this stems primarily from two key factors: (a) the way training is performed, and (b) the fact that peaBrain retains sequence information. Elastic net models only leverage the SNP dosage to predict expression in an individual. As highlighted above and in **Methods**, as peaBrain leverages differences between individuals (rather than prediction from the sequence directly, see **Methods** for implementation details), it is trained on a combinatorically larger dataset ($\binom{94}{2}$ = 4371 pairs of individuals). This allows peaBrain to learn more meaningful sequence information from a smaller number of individuals (assuming no relatedness). As a consequence, this training approach is prohibitively expensive with a larger number of individuals. However, by leveraging differences in the sequences between individuals to predict differences in expression (i.e., the sequence arrays are subtracted from each other), the resultant "difference sequence" captures how shifted the two sequences are and the differences in alleles. Thus, the distance information encoded implicitly by modelling the sequence is important to peaBrain performance. Without the sequence difference, peaBrain would simply be modelling SNP dosage (i.e., conceptually no different from existing [linear] models) and that is not sufficient for prediction of putatively functional variants in a relatively small dataset as observed.

Having established the predictive ability of peaBrain, we were interested in whether we can use the best-performing peaBrain models to measure the impact of single variants, compared to DeepSEA log fold change (logFC) estimates and experimental log skew estimates from MPRA and BiT-STARR-seq (**Task F**). For all "captured" genes (n = 113), we selected all variants identified as significant eQTLs in the GTEx v6p univariate eQTL analysis (n = 16,019 variants; **see Methods**) and replicated the analysis with the Geuvadis dataset [42] (n = 17,279 variants for the EU population and n = 1601 variants for the YRI [Yoruba from Ibadan, Nigeria] population). Unlike GTEx v7, eQTLs from GTEx v6p were derived from genotyping arrays (Illumina OMNI 5M or 2.5M) and thus did not include WGS used to train the peaBrain model. The Geuvdais dataset (both EU and YRI populations) were eQTLs derived from WGS from a set of non-overlapping subjects. For each eQTL (including indels), we created pairs of artificial sequences that only differed at the corresponding snp/indel position and predicted the difference in expression between the alternate and reference alleles from the difference between the two artificial sequences. (We used only a single model of the those several trained during cross-validation for simplicity, but incorporating results from additional models may improve results; see **Methods**.)

peaBrain predictions significantly and positively correlated with the univariate eQTL coefficients from the GTEx analysis (Spearman's rho = 0.09; p = 3.02 x10$^{-32}$; **S2 Fig**), from the EU-Geuvadis analysis (rho = 0.10; p = 9.60 x10$^{-38}$; **S3 Fig**), and from the YRI-Geuvadis analysis (rho = 0.18; p = 8.64 x10$^{-13}$; **S4 Fig**). Results were consistent across all datasets when we relaxed our heritability to include genes with GCTA p < 0.05: Spearman's rho equal to 0.08 (p = 3.02 x10$^{-25}$), 0.07 (p = 3.65 x10$^{-25}$), and 0.18 (p = 1.68 x10$^{-13}$) for the GTEx, EU-Geuvadis, and YRI-Geuvdais univariately-significant eQTLs, respectively. Alongside this positive correlation, we observed that many variants with large coefficients across all three datasets had small peaBrain predictions (**S2**–**S4 Figs**). This shrinkage in peaBrain estimates is consistent with the appreciable LD between eQTL-associated variants at any given locus, such that only a subset of significantly-associated variants are actually functional. Whilst univariate linear models are not capable of distinguishing between functional versus "hitchhiker" variants, the joint modelling of the input sequence in Stage 2 peaBrain allows a more direct assessment of the function of each variant. In comparison, we noted that the MPRA log skew estimates did not correlate

with the univariate eQTL coefficients in the GTEx (rho = 0.014; p = 0.60), the EU-Geuvadis (rho = -0.018; p = 0.58), or the YRI- Geuvadis (rho = 0.011; p = 0.82) eQTL analyses. The BiT-STARR-seq log skew estimates also did not correlate with the univariate eQTL coefficients in the GTEx (rho = -0.031; p = 0.50), the EU-Geuvadis (rho = 0.035; p = 0.44), or the YRI-Geuvadis (rho = -0.112; p = 0.32) eQTL analyses. Both the MPRA and BiT-STARR-seq experimental assays were run in lymphoblastoid cell lines. Similarly, for GM12878 (LCL) annotations, we noted that the maximum log fold changes from DeepSEA did not correlate with the magnitude of the eQTL coefficients in the GTEx analysis (rho = -0.001; p = 0.92), the EU-Geuvadis analysis (rho = -0.010; p = 0.18), and the YRI-Geuvadis (rho = 0.016; p = 0.52). The DeepSEA results suggest that simply predicting chromatin effects and TFBS is not sufficient for predicting the transcriptomic consequences of sequence variation for this set of selected genes. This is consistent with experimental evidence suggesting that most genetic variants in DNase footprinted (and other annotated) regulatory regions are silent [43], and the fact that histone modifications (e.g. methylation) alone are frequently insufficient to transcriptionally perturb promoters [44]. However, it is important to note that DeepSEA was trained on the reference genome (i.e., not exposed to genotype variation), while Stage 2 peaBrain was trained to predict differences in expression as a function of differences in sequence between any pair of individuals. Finally, and most importantly, we did not have any variant-level filters for any of the methods (e.g., using a p-value threshold for the experimental assays or any significance cut-off for the peaBrain estimates); thus, our comparison was not biased towards any method and assessed the utility of the method estimate across the range of variant effects.

Next, we sought to evaluate the performance of peaBrain at identifying putatively-functional eQTLs against empirical data from MPRA, BiT-STARR-seq, and HiDRA (**Task G**). Like MPRA, HiDRA is an extension of the classical reporter gene assay, adapted for sequence constructs derived from accessible DNA regions via ATAC-seq [26]; MPRAs leverage shorter synthesized DNA sequences [45]. BiT-STARR-seq is an extension of self-transcribing active regulatory region sequencing (STARR-seq), which like HiDRA involves fragmenting the genome and cloning fragments 3' of a reporter gene. We considered whether variants with the larger estimated effects from each of the three experimental approaches and peaBrain were preferentially located in sequences with known functional relevance (e.g. accessible DNA or transcriptionally active chromatin) and depleted from quiescent or repressed regions. The sequence annotations were derived from the Roadmap's GM12878 lymphoblastoid cell line 15-state ChromHMM model; the same GM12878 cell line was also used for both experimental assays (MPRA and HiDRA). BiT-STARR-seq was also performed in a lymphoblastoid cell line, but the exact cell line was not specified [25]. For each chromatin annotation, we assessed significance using a simple logistic model after rank-transformation of all estimates to normality (to ensure coefficients were comparable; see **Methods**). (It is important to note that our analyses are limited to the utility of these approaches in identifying and prioritizing putatively functional eQTLs; all of these experimental and *in silico* assays have other applications beyond functional eQTL discovery.) The coefficient of the model corresponded to the extent to which each approach was predictive of chromatin states/accessibility. More concretely, the logistic coefficients give the change in the log odds of the annotation overlap for a one-unit increase in the normalized score.

For peaBrain, as opposed to analysing the consequences of all possible variants/indels within 1Mbps input sequences for the "captured" 113 genes (which is computationally expensive), we focussed our analysis on all 23,595 univariately-significant eQTLs (from either the GTEx or Geuvadis datasets). We noted that variants with higher peaBrain estimates were significantly enriched in DNase accessible sites and transcriptionally active regions, and significantly depleted from heterochromatin and repressed sequences (**Table 2**). In contrast, the

**Table 2. In the 113 "captured" genes, eQTL variants with higher peaBrain estimates (i.e. more likely to be functional and with larger predicted transcriptomic impact) tended to fall in DNase accessible sites and transcriptionally active regions, and were similarly depleted from quiescent and repressed sequences (Task G).**

| | peaBrain | MPRA log-skew | | HiDRA | | BiT-STARR-seq log-skew | |
|---|---|---|---|---|---|---|---|
| | | all | shared | all | shared | all | shared |
| DNase accessibility | **0.12 ($3.76 \times 10^{-5}$)** | 0.01 (0.463) | 0.21 ($2.64 \times 10^{-2}$) | 0.01 (0.352) | -0.05 (0.732) | $-0.05\ (2.92 \times 10^{-13})$ | 0.09 (0.438) |
| TssA | **0.32 ($< 2 \times 10^{-16}$)** | 0.03 (0.218) | 0.25 ($2.05 \times 10^{-2}$) | -0.02 (0.140) | 0.01 (0.934) | 0.00 (0.965) | 0.09 (0.360) |
| TssAFlnk | 0.10 ($1.55 \times 10^{-2}$) | 0.06 ($4.72 \times 10^{-2}$) | 0.29 ($2.17 \times 10^{-2}$) | 0.03 ($4.01 \times 10^{-2}$) | 0.61 ($4.23 \times 10^{-3}$) | -0.02 ($1.28 \times 10^{-2}$) | 0.03 (0.788) |
| TxFlnk | -0.13 ($7.69 \times 10^{-2}$) | -0.01 (0.81) | -0.16 (0.672) | 0.08 (0.141) | 1.61 ($5.59 \times 10^{-2}$) | -0.02 (0.424) | -0.06 (0.787) |
| Tx | -0.02 (0.297) | -0.04 ($4.91 \times 10^{-2}$) | -0.05 (0.471) | -0.14 ($6.13 \times 10^{-3}$) | -0.80 ($9.58 \times 10^{-2}$) | 0.01 (0.571) | -0.07 (0.325) |
| TxWk | 0.04 ($1.42 \times 10^{-2}$) | -0.01 (0.662) | -0.01 (0.923) | 0.01 (0.630) | 0.48 (0.136) | **0.03 ($4.66 \times 10^{-4}$)** | 0.00 (0.973) |
| EnhG | 0.04 (0.547) | -0.12 ($6.97 \times 10^{-3}$) | -0.26 (0.167) | 0.04 (0.529) | -0.91 ($4.35 \times 10^{-2}$) | -0.05 ($1.85 \times 10^{-2}$) | -0.21 (0.349) |
| Enh | 0.03 (0.345) | 0.01 (0.693) | -0.04 (0.757) | 0.00 (0.910) | -0.71 ($3.23 \times 10^{-2}$) | $-0.03\ (1.39 \times 10^{-3})$ | 0.03 (0.872) |
| ZNFRpts | -0.07 (0.176) | 0.05 (0.310) | 0.03 (0.858) | -0.01 (0.910) | -0.06 (0.869) | -0.01 (0.724) | 0.16 (0.137) |
| Het | $-0.14\ (3.00 \times 10^{-7})$ | 0.04 (0.223) | -0.22 (0.169) | 0.05 (0.303) | -0.17 (0.816) | 0.01 (0.761) | 0.24 (0.101) |
| TssBiv | 0.66 (0.14) | -0.04 (0.881) | 1.10 (0.277) | -0.01 (0.958) | NA | -0.05 (0.381) | -0.18 (0.812) |
| BivFlnk | 0.19 (0.549) | -0.22 (0.293) | -0.52 (0.605) | -0.24 ($9.80 \times 10^{-3}$) | 0.15 (0.839) | -0.05 (0.356) | -0.19 (0.800) |
| EnhBiv | 0.40 (0.117) | 0.08 (0.701) | -0.46 (0.426) | 0.13 (0.306) | NA | -0.05 (0.302) | NA |
| ReprPC | 0.20 (0.129) | -0.21 ($3.80 \times 10^{-2}$) | NA | -0.05 (0.653) | -0.81 (0.440) | -0.03 (0.230) | -0.26 (0.735) |
| ReprPCWk | -0.25 ($< 2 \times 10^{-16}$) | -0.033 (0.133) | NA | -0.05 ($3.73 \times 10^{-2}$) | -1.00 ($6.80 \times 10^{-2}$) | -0.02 ($2.45 \times 10^{-2}$) | -0.19 (0.159) |
| Quies | 0.01 (0.342) | 0.02 (0.192) | -0.02 (0.667) | 0.00 (0.718) | -0.13 (0.564) | **0.02 ($4.22 \times 10^{-4}$)** | -0.01 (0.896) |

This trend was not observed for variants with large MPRA or BiT-STARR-seq log skew magnitudes, irrespective of whether we assessed performance on all variants on the platform or limited to univariately-significant eQTLs for the 113 "captured" genes. HiDRA performed better than MPRA and BiT-STARR-seq when using all variants assessed on the assay (all; n = 32,906 variants); performance further dropped when limited to the variants present in the peaBrain analysis (shared; n = 199). Point estimates were derived from fitting a simple logistic model with the scores from each assay rank-transformed to normality (i.e. model coefficients are directly comparable). Nominal p-value is presented in parentheses, but only entries that are significant after Bonferroni correction are shown in bold. (Green denoting enrichment; orange denoting depletion.) It is important to note that we did not filter based on the significance of the estimate for any of the methods (see **Main Text**). By comparing across all variants (without any significance filtering), we are able to show that peaBrain predictions from a single gene model are more informative (across the entire range of variant effects) than allelic log skew estimates from any of the experimental assays.

magnitudes of the MPRA log skew estimates were not significantly associated with any chromatin state or accessibility annotation after Bonferroni correction (**Table 2**). This absence of enrichment/depletion was consistent whether we analysed all variants assessed on the platform (n = 26,986 variants after excluding those with no match in Ensembl's VEP database; see **Methods**) or limited our analysis to the subset of variants also present in the peaBrain analysis (n = 1589 MPRA variants; i.e. univariately-significant eQTLs for the 113 "captured" genes). It is important to note that variants assessed on the MPRA platform were already selected, in part, because their eQTL status in the Geuvadis dataset; that is, excluding negative controls and LD-based selection, all variants assessed on the MPRA assays were univariately-significant eQTLs.

Similarly, variants with high magnitudes of the BiT-STARR-seq log skew estimates were not significantly enriched in transcriptionally active chromatin (or depleted from repressed/quiescent intervals), irrespective of whether we assessed performance on all variants assessed on the platform (n = 43,494) or limited to univariately-significant eQTLs for the 113 "captured" genes (n = 621). Using nominal p-value thresholds, HiDRA performed better than either MPRA or BiT-STARR-seq when looking at all variants assessed on the platform (n = 32,906 variants), but no annotation reached significance after multiple testing correction. Even when limited to the variants present in the peaBrain analysis (n = 199 univariately-significant eQTLs for the 113 "captured" genes), no significant enrichment or depletion was discovered for any annotation.

For all four methods, we did not apply any (significance) filter at the variant-level; that is, to ensure a fair comparison between all four methods, we did not select significantly active variants/fragments. Selecting the subset of variants significant for each method (e.g. using DESeq2 for HiDRA, QuSAR-MPRA for MPRA/BiT-STARR-seq, or a simple one-sample t-test across the peaBrain model repeats) would improve the results for the corresponding method (potentially biasing the test). It is important to note that we can generate confidence intervals/test-statistics for peaBrain estimates by assessing the prediction in each of the model replicates (trained and tested on different subsets of individuals); an idea conceptually similar to biological replicates in the experimental assays. However, the performance of a single cross-validated peaBrain model was deemed sufficient and thus, this assessment was not conducted. We should also note that the authors of the three experimental assays have convincingly shown that the methods, when limited to active fragments or significant variants (specific to each method), are able to identify functional variants enriched in transcriptionally active regions and depleted from heterochromatin [24–26]. However, this enrichment/depletion is limited to the subset of variants labelled as significant by the respective methods, i.e. the allelic log skew estimates are not insightful outside this limited subset. By comparing across all variants (without any significance filtering), we are able to show that peaBrain predictions from a single gene model are more informative (across the entire range of variant effects) than allelic log skew estimates from any of the experimental assays. In other words, peaBrain estimates can side step the noise inherent in assessing variant impact with experimental assays–suggesting that framework is useful in identifying putatively functional variants.

Having established that variants with higher peaBrain estimates are enriched in transcriptionally active chromatin (irrespective of any variant-level filtering), we sought to subsequently evaluate the four aforementioned methods on a more granular level using RegulomeDB [46] (**Task H**). The chromatin states and DNA accessibility assessed in **Task G** are only coarse indicators of variant function. RegulomeDB annotates variants in intergenic regions with known and predicted regulatory elements and categorizes each variant based on the evidence supporting regulatory function of the variant [46]. As RegulomeDB contains annotations from multiple tissues, we selected variants with well-established regulatory function in the GM12878 cell line ("Category 1"), which includes variants matched to known TF binding with matched TF motif and matched DNase footprint. For peaBrain and the three experimental assays (MPRA, BiT-STARR-seq, and HiDRA), we assessed significance using a simple logistic model after rank-transformation of all method estimates to normality (to ensure coefficients were comparable; see **Methods**). Similar to **Task G** (with chromatin states and accessibility), the coefficient of the model corresponded to the extent that each approach was predictive of variants with established regulatory function. In other words, larger coefficients indicate that the method is better able to delineate established regulatory variants from variants with minimal evidence for regulatory function. We note that only peaBrain had a significant and positive coefficient; with larger peaBrain estimates indicating variants with well-established and stronger evidence for predicted regulatory function (coefficient = 0.15 [0.04, 0.28]; **Table 3**). None of the three experimental assays had significantly positive coefficients (for all variants tested on the respective platforms and limited to the subset of eQTLs for the 113 "captured" genes). Overall, on the post-selective 113 genes, **Tasks F-H** suggest that the modelling undertaken by Stage 2 peaBrain (derived from sequence data alone) detects functional architecture that is not readily accessible with the latest high-throughput empirical approaches.

## Discussion

Here, we have introduced a two-stage computational framework for predicting the transcriptomic consequences of non-coding variation. Using Stage 1 class-C (tissue-specific annotated)

**Table 3. In the 113 "captured" genes, peaBrain estimates can significantly delineate variants with established regulatory function (Task H).**

| Method | Variants | Logistic Coefficient | L Bound | U Bound | p-value |
|--------|----------|---------------------|---------|---------|---------|
| peaBrain | | **0.16** | **0.04** | **0.28** | **7.99 x10⁻³** |
| MPRA | all | 0.10 | -0.002 | 0.19 | 5.46 x10⁻² |
| | shared | 0.25 | -0.06 | 0.56 | 0.119 |
| BiT-STARR-seq | all | -0.03 | -0.11 | 0.04 | 0.371 |
| | shared | 0.09 | -0.16 | 0.34 | 0.461 |
| HiDRA | all | 0.09 | -0.02 | 0.20 | 0.107 |
| | shared | 0.11 | -0.30 | 0.51 | 0.603 |

The log-skew estimates from the experimental assays, both across all variants assessed on each platform ("all") and limited to eQTLs for the 113 "captured" genes ("shared"), are uninformative. Both the peaBrain estimates and log-skew for the experimental assays were rank-transformed to normality to facilitate comparison between the methods. The bounds for the 95% confidence interval, obtained by profiling the likelihood function, are tabulated, with significant coefficients denoted in bold. **Abbreviations:** L, lower; U, upper.

models, we observed that the majority of variance (>50%) in the mean abundance of genes across most GTEx tissues is encoded in the annotated 4kbps core promoter sequences. Thus, the difference in mean abundance between genes appears to be largely encoded in invariant differences between core promoter elements and the interacting tissue-specific regulatory factors encoded in the model weights, rather than a consequence of transcriptional regulation by more distal sequences or non-transcriptional downstream regulation (e.g. silencing by small non-coding RNAs). Furthermore, we note that the average expression of all genes in a single tissue and the reference genome is sufficient to learn both TFBS and allele-specific binding (see **S1 Text**). Taken together, this is broadly consistent with anecdotal experimental evidence [31] and suggests that non-transcriptional downstream processes play a secondary role in regulating mean expression.

The predictive ability of Stage 1 peaBrain models allowed us to calculate a non-coding impact score for all genomic positions in the core promoter sequences, a useful metric for analysis of both common rare variants. Unlike other non-coding metrics that incorporate external consequence annotations (e.g. from Ensembl's variant effect predictor [VEP], ClinVar, and other curated databases), peaBrain impact score is derived directly from predicting expression and does not depend on curated variant annotations. The tissue-specific nature of the peaBrain impact score is useful for identifying putatively functional tissues underlying GWAS signal for complex traits, which are not readily accessible through current methods that rely on eQTL-GWAS-hit co-localization.

To incorporate the consequences of individual variation on gene abundance in Stage 2 of the framework, we extended the Stage 1 model to capture a 1Mbps window, a balance between computational tractability and biological "signal". Unlike Stage 1 models, Stage 2 peaBrain leverages differences in the sequences between individuals to predict differences in expression (rather than prediction from the sequence directly, see **Methods** for implementation details); that is, the sequence arrays are subtracted from each other and the resultant "difference sequence" captures how shifted the two sequences are and the differences in alleles. Without the sequence, peaBrain would simply be modelling SNP dosage (i.e. conceptually no different from existing [linear] models) and that is not sufficient for prediction of putatively functional variants as observed. Thus, the distance information encoded implicitly by modelling the sequence appears important to peaBrain performance. However, peaBrain is a blackbox approach and we must be cautious in attempting to elucidate scientific rationales for the apparent improved performance. Existing methods for peering into "black-box" approaches

are not particularly useful for peaBrain as it leverages differences between individual sequences aligned to the annotated TSS, rather than conventional (reference) sequences, that are modelled with "conjoined" neural networks (see **S7 Table**). In other words, we cannot readily extract meaningful motif sequences from the input data. Reconstruction of the individual sequences to generate the difference input required that we use a quality controlled VCF to reconstruct individual sequences (see **Methods**), as opposed to directly using the originally "noisy" sequence reads. However, by leveraging differences in "TSS-aligned" sequences, peaBrain learns to map differences at each genomic position of an individual (relative to the fixed TSS landmark) to predict difference in expression. The advantage of this approach is that peaBrain must learn to pinpoint important features regardless of where they occur in the sequence and that may eschew the overfitting concern associated with *a priori* identification of eQTLs. Importantly, Stage 2 peaBrain does not directly depend on eQTLs/variation dosage, but rather focusses on how differences at each genomic position (because of differences in alleles or because of shifts due to upstream/downstream indels) perturb expression.

At this conjecture, it is important to note that, unlike many methods with similar conceptual origins, peaBrain was not designed with the sole intent of predicting gene expression abundance. Rather, one of the primary goals of Stage 2 peaBrain models is identifying putatively functional eQTLs. As a first approximation, we note that peaBrain variant effect estimates positively and significantly correlate with the coefficients from the univariate eQTL analysis on the post-selective 113 "captured" genes. In contrast, MPRA and BiT-STARR-seq allelic log skew estimates did not correlate with the corresponding univariate eQTL coefficients. Furthermore, variants with large peaBrain estimates were significantly enriched in DNase-accessible DNA and transcriptionally active chromatin, and depleted from quiescent and repressed states. Log skew estimates, for both MPRA and BiT-STARR-seq for variants were uninformative of chromatin state for the subset of variants investigated. The poor performance of MPRA may reflect the fact that it is an episomal assay so variants are not being assessed in their regular chromatin context. Variants with large HiDRA estimates were nominally enriched in transcriptionally active regions, but did not reach significance after Bonferroni correction. Notably, however, both the MPRA and HiDRA assays were performed in the GM12878 cell line from which the chromatin and DNA accessibility annotations were also derived, i.e. there is a possibility that the results for the experimental assays are biased overestimates of true performance. It is important to note that when limited to the subset of significant variants (as labelled by each method), the experimental assays can identify regulatory variants enriched in transcriptionally active chromatin. The log-skew estimates from any of the three experimental assays, however, cannot delineate functional variants outside this limited "significant" subset. In **Tasks F-H**, by comparing across all variants (without any significance filtering), we show that peaBrain variant predictions are more informative (across the entire range of variant effects) than allelic log skew estimates from the experimental assays. More concretely, as described above, peaBrain estimates can side step the noise inherent in variant-level measurements using *in vitro* empirical assays.

Our initial explorations, using peaBrain, have demonstrated that regulatory activity up to 1Mbp away from TSS can predict differences between individuals for a subset of genes. The next stage of this work is design more efficient and computationally inexpensive models. Transformers, which are at the heart of DeepMind's Enformer, provide one option: modelling relationships between genomic positions regardless of their respective position. However, as we have shown, the position of the variants also encodes valuable observation. As such, transformers provide only part of the solution; new methods must both model the influence of enhancers and distant regulatory regions as well as retain their positional information (which is necessary to identify person-to-person differences).

As with other deep learning approaches, there are limitations to peaBrain analysis; notably, that despite our best efforts for the rigorous quality control, unbiased error estimation, and model regularization, there may be some information that is biasing performance results in an intricate way (i.e. the generic problem of using black-box neural network models). To mitigate the risk of bias, we implemented dropout regularization, out-of-sample testing on unrelated individuals (after conservatively filtering for cryptic relatedness), comparison with high throughput assays (such as MPRAs and HiDRA), and validation using chromatin, TF-binding, and DNA accessibility annotations. However, without an explicit model, there is always a possibility for bias. For peaBrain, the ability of the Stage 2 analyses to identify putatively functional variants that are enriched in transcriptionally active chromatin and depleted from heterochromatin/repressed sequences is encouraging evidence of model generalizability. Similarly, the correlation of the impact scores from Stage 1 analyses with evolutionary constraint and their utility in predicting disease-associated mutations and allele-specific binding sites further underscores the true performance of peaBrain framework.

Altogether, the results from the Stage 1 and Stage 2 of the peaBrain framework suggest that models for understanding the effects of non-coding variation on RNA abundance (and possibly more complex traits) can be built by relying more on automated machine learning, rather than hand-designed or selected features. Furthermore, the results highlight the variant sensitivity of the Stage 2 peaBrain model and its ability to identify putatively functional variants underlying cis-eQTL signals. More generally, peaBrain's performance in predicting mean abundance and individual variation further implicates the importance of the invariant genomic context and distance to the annotated TSS for interpreting the effects of non-coding variation in a tissue-specific manner.

## Methods

**RPKM and gene count** data, for Stage 1 and Stage 2 peaBrain models, was downloaded from GTEx (v7; see URLS in S4 Text) [28]. To prepare the data for Stage 1 of peaBrain, the mean abundance of each gene was obtained by averaging the RPKM across all subjects. The values were then rank transformed to normality using the rntransfrom function from GenABEL v1.8–0. The GRCh37 (hg19) reference genome was downloaded from UCSC [27]. We used the default Ensembl gene definitions to define gene borders; an Ensembl gene is defined as the collection of all spliced transcripts with overlapping coding sequences but excluding manually annotated readthrough genes. The gene start and end coordinates (from which the core promoter sequences are defined) correspond to the outermost transcript start and end coordinates. We accounted for gene strand-ness while extracting the core promoter sequences; start coordinates corresponding to the TSS for genes on the positive strand and the end coordinate corresponding to the TSS for genes on the negative strand. We further limited our analysis to protein-coding genes (n = 19,820 genes) and to autosomal chromosomes for simplicity. For all Stage 1 models, the DNA promoter sequence for each gene was one-hot encoded (also known as a one-of-k scheme); each letter represented as separate channel. One-hot encoding is a technique commonly used in natural language processing to encode categorical integer features with each channel indicating the presence (1) or absence (0) of the corresponding DNA letter. Processed genomic annotations and epigenetic markers were obtained from the LDSC [29] (see **URLS in S4 Text**) and similarly processed. For Stage 1 class B models and using the LDSC annotations, we incorporated an additional 28 channels of binary sequences for each basepair, that are not specific to any cell type or tissue, highlighting: coding basepairs, conserved sites [47], CTCF sites, DGF peaks [2], DHS peaks [3], enhancers [4,48], fetal DHS peaks [3], H3K27ac peaks [5,30], H3K4me1 peaks [3], H3K3me3 peaks [3], H3K9ac peaks [3], introns

[27], promoters [27], promoter flanking sequences [4], repressed sites [4], super enhancers [5], transcription factor binding sites (TFBS) [2], transcribed sequences [4], TSS [4], untranslated 3' regions (UTR3) [27], untranslated 3' regions (UTR5) [27], and weak enhancers [4]. Stage 1 class C models included additional binary channels, corresponding to the consolidated epigenomes from Roadmap (see **URLS in S4 Text**), as described in the main text. Transcription factor processed ChIP-seq data were also downloaded from the gene transcription regulation database (GTRD v17.04; see **URLS in S4 Text**). GTRD is a database of human transcription factor binding sites identified from ChIP-seq experiments and uniformly processed. As described in **S1 Text**, for a subset of Stage 1 models, transcription factor binding sites identified using four different peak callers (MACS, SISSR, GEM and PICS) and clusters of peaks for each method (defined as overlapping peak called using the protocol, but in different tissues or under different conditions) were included as separate binary channels.

**Stage 1 peaBrain model** was constructed using Theano 0.9.0 and Lasagne 0.1. For a single tissue, peaBrain takes in the core promoter sequence as input and predicts the normalised mean abundance of the corresponding gene (**Fig 1**). The core promoter sequence was determined by varying the length of the promoter sequence (± 1kbps, ± 2kbps, and ± 3kbps). As highlighted in **S1 Fig**, ± 2kbps (i.e. the 4kbps core promoter sequence) was the optimal length for predictive ability as assessed using a 10-fold cross-validation scheme. The input sequence is a 1D vector with 4 channels encoding the DNA sequence and when appropriate, additional channels as binary representations of various genomic annotations and epigenetic markers (described above). The Stage 1 peaBrain model is a series of 1D convolutions and max pooling layers (**S6 Table**). In practice, a 1D convolution is implemented as a 2D convolution with width set to 1 (effectively dropping the unused dimension). Each convolutional layer was set with 11 filters of size 5 and a leaky rectify non-linearity activation function. The leaky rectify activation function for all convolutional layers has a nonzero gradient for negative input, which is useful for convergence [49]:

$$f(x) = \begin{cases} x & \text{if } x > 0 \\ 0.01x & \text{if } x \leq 0 \end{cases}$$

The 0.01 corresponds to the "leakiness" of the activation function, with larger values denoting increased "leakiness". The input to the first convolutional layer is 4000 x 1 x r sequence, where 4000 corresponds to the length of the core promoter sequence and r denotes the number of channels (minimum of 4 DNA letter channels). The first convolutional layer has 11 filters (or equivalently, kernels) of size 5 x 1 x 11, where 5 denotes the sequence length of the filter and 11 denotes the number of channels for that filter. The output of each filter is a locally connected structure, convolved with the sequence, to produce 11 feature maps that are then max pooled with the output of other filters from the layer, before serving as input for the subsequent layer. Prior to the penultimate embedding layer (from which we extract the continuous vector gene representations), we placed a dropout layer with p = 0.5 of setting values to zero. The dropout layer is a regularizer that randomly zeros input values (i.e. randomly dropping units and their connections), limiting co-adaptation and improving model generalizability [50,51]. The number of units in the penultimate embedding layer determines the size (or the number of components) in the vector and was set to 1001. The last layer is a single output neuron that outputs the mean abundance of the corresponding gene (for which the promoter was input). The last two dense layers (including the final output neuron) have linear activations, ensuring that the normalized mean abundance is a linear combination of the embedding components or equivalently, the neural activations of the penultimate layer. The objective was defined using the mean squared difference (between predictions and observed mean

abundances) and model weights were updated using Adam with the learning rate = 0.001, beta1 = 0.9, beta2 = 0.999, and epsilon = 1 x10$^{-8}$. Adam is an algorithm for gradient-based optimization of (stochastic) objective functions [52]; beta1 corresponds to the exponential decay rate for the first moment estimates and beta2 is the decay rate for the second moment estimates. The model was trained for a minimum of 100 epochs, before exiting early using a validation set (defined as 10% of the training). As is typical in neural networks, the number of layers and other explicitly-specified model variables, above, are referred to as hyperparameters; they are variables that set prior to optimization of the models parameters.

**Pre-processing for heritability and variant-sensitive regression** for the Stage 2 peaBrain model was performed as recommended by the authors of QTLTools [53]. For each tissue, we selected genes with non-zero RPKM values for at least 50% of samples. Per gene, RPKM values were residualised using linear regression to account for autolysis score, date of nucleic acid isolation, date of genotype isolation, RIN, total ischemic time, time spent in paxgene fixative, sex, age, Hardy score, interval of onset to death for last underlying cause, number of hours in refrigeration, ischemic time, temperature, donor status (post-mortem, surgical or organ donor), three genotype PCs and enough expression matrix PCs to explain 55% of the variance (to account for unexplained technical and biological variance). The residuals were then rank-transformed to normality using GenABEL's rntransform function. As with Stage 1 peaBrain analyses, we further limited our analysis to protein-coding genes (number of genes differed between tissues) and to autosomal chromosomes for simplicity. For each protein-encoding gene on an autosomal chromosome, we defined the input sequence as 0.5 Mbps upstream and 0.5Mbps downstream of the TSS using default GRCh37 Ensembl gene definitions (total 1Mbps centred on the TSS). As highlighted in the main text and, the 1Mbps was selected as a balance between computational tractability and biological relevance. Increasing the length of the input sequence beyond the 1Mbps increases both the compute time and memory footprint. Importantly, our symmetric 1Mbps window likely contained >95% of cis-eQTLs; in the GTEx dataset, the 95$^{th}$ percentile for absolute distance of cis-eQTLs from their target transcript TSS was 441,698bps [28]. Incidentally, the 1Mb interval have also used by other approaches in imputing RNA expression from genotype (namely, TWAS) [9]. Using the unphased whole genome sequencing GTEx data, we reconstructed the individual's sequence from the quality controlled VCF. In other words, we generated the individual variation by substituting each individual's non-reference alleles into the reference sequence. Variants in the WGS GTEx VCF were quality controlled by GTEx LDACC at the Broad Institute. As stated in the corresponding README file, quality control was conducted using GATK, Hail, and PLINK. Notably, a variant was removed if it didn't "pass Variant Quality Score Recalibration (VQSR), had low Inbreeding Coefficient or low Quality Score, was within a Low Complexity Region (LCR), became monomorphic after applying genotype quality score (GQ) <20 or allele balance (AB) >0.8 or AB<0.2 filters or assigning male heterozygous calls in chrX nonPAR regions to missing, had missingness rate > = 15%, did not pass Hardy-Weinberg Equilibrium testing in African American or European subpopulations for autosomes or in European females for chr X, showed significant association with sequencing technology or library construction batch, or showed significant non-random missing of reference alleles." [28] For each individual, we generated two copies of the gene 1Mbps input sequence; phasing did not matter as the sequences were combined prior to modelling.

**Stage 2 peaBrain models for heritability analysis** and variant-sensitive prediction were similarly constructed as described for the Stage 1 models. Stage 2 models, however, are three separate convolutional neural networks, connected by a dense fully-connected layer prior to the output neuron (**S7 Table**). The input 1Mbps sequence is split into three inputs: 0.48Mbps upstream, 4kbps core promoter, and 0.48Mbps downstream sequences. The 4kbp core

promoter is the input to a CNN with identical structure and hyperparameters as described for Stage 1 peaBrain model (described above in detail). The upstream and downstream sequences are input to networks with identical architecture, but different pooling hyperparameters: a pool size of 100 for the first pooling layer, 50 for the second, and 10 for the last. Number of filters was consistent between all networks (n = 11). The fully connected output from each sequence is concatenated, before one penultimate fully-connected layer and a single output node. Unlike the Stage 1 peaBrain models, Stage 2 models are trained to predict the differences between individuals (rather than direct prediction of expression). As humans are diploid, for each individual, the input sequence was the sum of the one-hot encoding of each of the 1Mbps sequences corresponding to the "maternal" and "paternal" sequences; phasing did not matter because the sum was consistent. A separate Stage 2 model was constructed for each gene with significant non-zero heritability (see **text**). For any pair of individuals, A and B, the input sequence was defined as the difference between the one-hot encoded sequences, with the corresponding output as the difference between the two individuals. We included both differences, (A–B) and (B–A), during training. After removing individuals with cryptic relatedness (see GCTA analysis below), the GTEx dataset was randomly split into train and test individuals (95% of subjects for training and 5% for testing), with the model trained on all the pairwise differences between train individuals and tested on all pairwise differences between test individuals. The training set was further sub-divided into training and validation sets, with the latter used to exit early after a minimum 100-epoch training. As described below, overall model performance was assessed using the cv-$r^2$ on five to ten random repetitions of 95/5 train/test splits; the number of repetitions was dependent on how quickly each model reached exit criteria.

**Elastic net (regularized linear) models.** We used an additive genetic model as our baseline comparison as described elsewhere [6]. Briefly, for each gene, an elastic net model was used to model expression (alpha = 0.5; selected to match PrediXcan [6]). As with peaBrain, the models were trained to predict the difference in expression as a function of the difference in dosages among the variants within the 1Mb input sequence (rather than the expression directly). For a linear model, this is no different from simply predicting the expression; the constant term in this case is expected to be close to 0. The lambda (regularization) parameter was 3-fold cross-validated on the training dataset, using cv.glmnet function from glmnet v2.0–10 [54].

**Model performance,** for all peaBrain and linear models, was assessed using the cross-validated-$r^2$ (cv-$r^2$), a classical machine learning metric to assessing performing of regression models (often just called $r^2$) [55]. cv-$r^2$ is defined as:

$$cv-r^2 \equiv 1 - \frac{\sum_i (y_i - f_i)^2}{\sum_i (y_i - \bar{y}_{test})^2}$$

where $f_i$ denotes the predicted value using the model fitted on the training data, $y_i$ denotes the true value for, and $\bar{y}_{test}$ is the mean value for all items in the test set. The denominator of the cv-$r^2$ is the total sum of squares (proportional to the variance of the data) and the numerator of the cv-$r^2$ is the explained sum of squares (also called the regression sum of squares). When the explained sum of squares (numerator) is larger than the total sum of squares (denominator), cv-$r^2$ is below zero and indicates the model does not have any predictive ability. Regression models with some predictive capacity have cv-$r^2$ values in the range (0, 1]. Stage 1 peaBrain model performance was assessed using 10-fold cross validation (10% of genes were withheld from the algorithm during training). Stage 2 peaBrain models and elastic net linear models were assessed using repeated random splits of 95% of subjects for training and 5% of subjects for testing. Individuals with cryptic relatedness were removed prior to the training/

test split, using GCTA grm-cutoff of 0.025 (see below). 5–10 random training/test splits were used to assess model performance; 95% confidence interval was estimated using the mean and standard error, assuming the distribution of cv-$r^2$ was normal.

**Tissue-specific peaBrain impact score** for any given genomic position, as described in **text**, was defined as the absolute difference in abundance between the original promoter sequence and a modified promoter sequence where all the sequence and epigenetic/genomic annotations for that site were set to zero. The impact score is proportional to the contribution of the genomic position to the average expression of the gene; genomic positions are readily mapped to genes by virtue of the promoter definitions. If the genomic position overlapped with the promoter of multiple genes, the maximum impact across all overlapping genes was taken. Tissue-specific peaBrain impact scores were compared to phylogenetic p-values (phyloP) using simple linear models (lm base function in R). As briefly described in the main text, phyloP are nucleotide conservation scores derived from multiple alignments of 99 vertebrate genomes to the human genome phyloP scores are based on an alignment and a model of neutral evolution [27]. A more positive value indicates conservation or slower evolution than expected; magnitude of the phyloP score corresponds to the -log p-values under the null hypothesis (i.e. neutral evolution). phyloP scores were downloaded from the UCSC genome browser (see **URLS in S4 Text**). To compare peaBrain to other non-coding metrics, a non-tissue-specific peaBrain score was used; defined as the average impact of each position across all tissues. Non-coding impact scores (combined annotation dependent depletion [CADD] v1.3, and Eigen v1.1) were downloaded from their respective webpages (see **URLS in S4 Text**). CADD is a single meta-score derived from analysis of multiple annotations for variants that survived natural selection, compared to simulated mutations [16]. Eigen is an unsupervised score that synthesizes a combination of functional annotations into one meta-score [17]. The non-coding somatic mutations used to assess metric performance were downloaded from the COSMIC v82 (see **URLS in S4 Text**). Allele frequency was derived from gnomAD release 170228. For each genomic position, we counted the number of overlapping somatic mutations. We further limited our analysis to COSMIC census genes (as a positive gene set); COSMIC census genes possess mutations that have been causally implicated in cancer. For each task used to compare the non-coding metrics (see text), a logistic model was used (fitted using the glm function in R, family = "binomial") with the allele frequency and phyloP as covariates. Allele frequency and evolutionary conservation scores were included to assess whether the non-coding impact score adds any additional information to the model, besides that derived from allele frequency or evolutionary constraint. Positions without a phyloP conservation score were excluded from model fitting. The confidence interval was obtained using the confint function (derived from profiling the likelihood function). For the analysis of recurrent somatic mutations, we were interested in the global performance of each metric at each autosomal chromosome and thus a simple model sufficed–isolating genes or promoters with "mutation hotspots" would require more sophisticated approaches to avoid false positives (e.g. it would be necessary to incorporate tumour type, the proportion of each tumour [sub]type, the background mutation rate at each position/tumour, and more technical variables such as sequence coverage). The published non-coding impact scores (CADD and Eigen) depend on curated non-coding annotations and indirectly predict transcriptomic consequences; that is, there is potential risk of overestimating the performance of these scores in the three tasks (see **Main Text**). Allele-specific binding site data and prediction scores for TF binding prediction algorithms were downloaded from the Supplemental Table appended to Wagih *et al.* [33] (see **URLS in S4 Text**). For the comparison between non-coding impact metrics, duplicate sites were filtered (selecting the one with lowest nominal p-value). For the analysis of causal tissues, we downloaded the summary statistics for the four lipid traits from the webpage of the Global

Lipids Genetics Consortium (see **URLS in** S4 Text). Local SNP-heritability for each trait was calculated using HESS (Heritability Estimation from Summary Statistics). The linear models of local heritability as a function of average peaBrain score per locus were fitted using the base function *lm* in R.

**Constrained GCTA heritability analyses.** We converted the GTEx whole genome sequencing VCF to PLINK binary bed file (using Plink v1.9). Using GCTA v1.24.4 [41], we calculated the genetic relationship matrix (GRM) from all the autosomal SNPs and excluded individuals with grm-cutoff of 0.025. GCTA was used to calculate heritability for similar methods, including predixcan [6] and TWAS [9]. For each gene, we subsequently limited the GRM to variants within the 1Mb input sequence (centred on the TSS) and performed constrained GCTA-GREML analysis. Genes with a significant non-zero heritability ($p < 0.01$) were included for subsequent analyses.

**Predicative ability of gene embeddings** was assessed using a 10-fold cross validation scheme. The hallmark curated gene sets were downloaded from Molecular Signatures Database v6.0 [56]. Hallmark gene sets represent an aggregation of many gene sets and are thought to represent coherent biological states or processes. For each set, genes were assigned a binary label (1 denoting membership). We subsequently trained a multi-layer perceptron classifier from scikit-learn v0.19.0, with three hidden layers (200, 100, and 50 neurons), to predict gene-set membership using the gene's embedding. We used a rectified linear unit function as the activation for our hidden layers, and lbfgs for weight optimization. Lbfgs is an optimizer that belongs to the family of quasi-Newton methods. Cosine similarity between any pair of embeddings was assessed using eponymous function from scikit-learn, defined as:

$$\text{similarity} \equiv \frac{X \cdot Y}{\|X\|\|Y\|}$$

where X and Y denote the embeddings for genes X and Y, respectively. Correlation between the RNA-seq arrays for genes X and Y were calculated using the base cor function in R.

**Correlation with univariate GTEx/Geuvadis eQTL analysis, DeepSEA, and MPRA & BiT-STARR-seq log skew estimates.** To calculate the effects of single variants, artificial sequences were constructed that differed only at the genomic position of the corresponding variant; with one sequence containing the reference (ref) allele and one sequence containing the alternate (alt) allele. As Stage 2 peaBrain model predicts the difference between two sequences, we used the (alt–ref) configuration to estimate an effect size for each variant. Univariate eQTL coefficients were obtained from the GTEx and Geuvadis datasets (see **URLS in** S4 Text). Significance of spearman (rank) correlation between the peaBrain estimate and eQTL coefficient was assessed using the cor.test function in R. Both the univariate eQTL analysis and peaBrain were obtained using expression data that was rank transformed to normality and thus are comparable in magnitude (despite slightly different pre-processing protocols). MPRA variant results were obtained from Supplemental Table 1 of Tewhey *et al.* [24] (see **URLS in** S4 Text); snp rs ids were translated to chromosome_position_ref_alt_build nomenclature using Ensembl's GRCh37 biomaRt and a simple python script. Any variant that intersected with the peaBrain final variant set was included in the analysis, that is, variants in the 1Mbps input sequence for genes/models with the 95% confidence interval for the cv-$r^2$ entirely above zero. The LogSkew.Comb column, corresponding to the log2 allelic skew from the combined MPRA LCL analysis (alt/ref), was used as the MPRA log skew estimate. The BiT-STARR-seq data was similarly processed (see **URLS in** S4 Text). As with the peaBrain and eQTL analysis, significance of the spearman (rank) correlation between each of the experimental assays allelic log skews and the univariate eQTL coefficients was assessed using the cor.test

function. To obtain the logFC for GM12878 annotations, a vcf file of the corresponding eQTLs was uploaded to the DeepSEA platform (see **URLS in S4 Text**).

**Comparison of peaBrain, MPRA, BiT-STARR-seq and HiDRA.**   The core 15-state model and DNAse accessibility annotations for the GM12878 EBV-transformed lymphoblastoid cell line (LCLs) were downloaded from the Roadmap project (see URLS in S4 Text). HiDRA data was downloaded from the GEO series GSE104001 (see URLS in S4 Text). HiDRA estimate was defined as the log fold change in average counts between the alternate and reference group (after normalizing for DNA count); direction did not matter as only the magnitude was used in this analysis. The MPRA and BiT-STARR-seq data was downloaded and pre-processed as described above. Notably, the chromatin states/DNA accessibility annotations, the HiDRA, and MPRA estimates were derived from the same cell line; that is, there is possibility of overestimating the performance of either method. For any given annotation, we assessed the predictive ability of the magnitude of the variant estimate (from any of the three approaches) to predict whether the variant overlapped with the annotation. The magnitude of the variant estimate for each approach (peaBrain, HiDRA, BiT-STARR-seq, and MPRA) corresponded to either the transcriptomic impact or activity of that variant. Only the absolute magnitude, after rank-transformation to normality, of each variant was used in modelling. For each approach and for each annotation, a logistic model was used (fitted using the glm function in R, family = "binomial") and the confidence interval was obtained using the confint function (derived from profiling the likelihood function). For the granular variant-level assessment, annotations were downloaded from RegulomeDB (dbSNP 141; see URLS in S4 Text).

## Supporting information

**S1 Table. Tabulated summary of coefficients of the linear function modelling phyloP conservation scores as a function of tissue-specific peaBrain noncoding impact metric.** Generally, across most tissues and chromosomes, the larger the impact a position has on the mean abundance of the gene (as indicated by a higher peaBrain impact metric), the more evolutionary conserved it is (i.e. a positive coefficient). The notable exception is the nucleus accumbens (basal ganglia), where the opposite trend is noted (negative coefficients; in bold). All coefficients are significant ($p < 10^{-16}$). The results were also consistent with the rank-normalized phyloP and peaBrain scores. **Abbreviations:** L, lower; U, upper.
(DOCX)

**S2 Table. Tabulated summary of all tasks used to assess peaBrain performance, for both Stage 1 and Stage 2 models.**
(DOCX)

**S3 Table. Tabulated p-values for the top five putatively functional tissues per trait (ranked in ascending order by p-value), as predicted by the peaBrain framework and the RTC (eQTL)-based methodology (Task D).** peaBrain p-values have been Bonferroni-corrected for multiple testing; results for all tissues are available in **S4 Table**. Nominal p-values are shown for the RTC (eQTL)-methodology; obtained from S8 Table of the corresponding manuscript [39]. Across all tested traits, the peaBrain framework identifies more relevant functional tissues per trait than the RTC-based method. **Abbreviations:** LDL, low-density lipoprotein; HDL, high-density lipoprotein; RTC, regulatory trait concordance.
(DOCX)

**S4 Table. Causal tissue profiles for all lipid traits.**
(XLSX)

**S5 Table. Performance metrics (confidence and point estimates for cv-r$^2$ from both classes of models) and estimated GCTA heritability for all genes with significant heritability (GCTA p < 0.01).**
(XLSX)

**S6 Table. Schematic of the Stage 1 peaBrain model.** The number of channels, r, is determined by the number of epigenetic and genomic annotations included in the model (minimum of 4 corresponding to the 4 DNA letter channels in class A models). The Stage 1 class B models have 32 channels, corresponding to 4 DNA sequence channels and 28 annotation channels (see **Methods** for details).
(DOCX)

**S7 Table. Schematic of the Stage 2 peaBrain model, which is composed of three separate networks connected by a dense layer prior to prediction.** Values in red denote layers with differing values between the three networks. The network for the centre split is identical to the Stage 1 peaBrain model for the core promoter region; the networks for the upstream and downstream splits are identical to the Stage 1 peaBrain model for distal sequences. Thus, Stage 2 peaBrain can be thought of as a consolidation of the separate Stage 1 models.
(DOCX)

**S1 Fig. Using the class-B peaBrain model for MuscleSkeletal (largest tissue by sample count in GTEx), the 4kbps promoter sequence (+/− 2kbps of annotated TSS) outperforms both 2kbps (+/− 1kbps) and 6kbps (+/− 3kbps) promoter sequences in predicting mean gene abundance.**
(PNG)

**S2 Fig. Scatter (*right*) and hexa-bin (*left*) plots of variant-expression effects as estimated in LCLs by peaBrain (limited to genes whose 95% confidence interval for the cv-r$^2$ is entirely above 0; n = 113 genes; Task F).** Each point corresponds to a variant that is univariately significant in the GTEx eQTL analysis (n = 16,019 eQTLs). The y-axis is the magnitude of the univariate GTEx eQTL coefficient for the corresponding variant. The correlation between the GTEx coefficient and the peaBrain prediction is positive and significant (Spearman's rho = 0.09; p = 3.02 x10$^{-32}$).
(PNG)

**S3 Fig. Scatter (*right*) and hexa-bin (*left*) plots of variant-expression effects as estimated in LCLs by peaBrain (limited to genes whose 95% confidence interval for the cv-r$^2$ is entirely above 0; n = 113 genes; Task F).** Each point corresponds to a variant that is univariately significant in the EU-Geuvadis eQTL analysis (n = 17,279 eQTLs). The y-axis is the magnitude of the univariate EU-Geuvadis eQTL coefficient for the corresponding variant. The correlation between the EU-Geuvadis coefficient and the peaBrain prediction is positive and significant (Spearman's rho = 0.10; p = 9.60 x10$^{-38}$).
(PNG)

**S4 Fig. Scatter (*right*) and hexa-bin (*left*) plots of variant-expression effects as estimated in LCLs by peaBrain (limited to genes whose 95% confidence interval for the cv-r$^2$ is entirely above 0; n = 113 genes; Task F).** Each point corresponds to a variant that is univariately significant in the YRI-Geuvadis eQTL analysis (n = 1601 eQTLs). The y-axis is the magnitude of the univariate YRI-Geuvadis eQTL coefficient for the corresponding variant. The correlation between the YRI-Geuvadis coefficient and the peaBrain prediction is positive and significant (Spearman's rho = 0.18; p = 8.64 x10$^{-13}$).
(PNG)

**S1 Text. Validation of peaBrain Stage 1 and its utility in diverse set of tasks.** The note also highlights how activations of the penultimate layer of the peaBrain model can be used as a continuous and compressed representation (i.e. embedding) of genes. These embeddings, or equivalently, neural activations, capture both the annotated DNA (input) and its additive contributions to tissue-specific abundance (output) in a compressed form amenable to downstream analyses (such as network-based analyses). These embeddings display interesting properties, including the encoding of correlation information and membership to pathways/curated gene sets. Importantly, these embeddings are in a linear space, such that the pairwise cosine similarity between these dense gene representations is proportional to the measured RNA-seq correlation between the gene pair. In other words, co-regulation and co-expression may be discovered by leveraging linear structure within the embeddings (e.g., adding embeddings of two genes to discover their co-expression with a third).
(DOCX)

**S2 Text. We demonstrate the utility of this peaBrain DNA reporter model in investigating the role of DNA and histone modifications in difficult-to-study processes, such as neural induction.** By modelling the transitions between consecutive neural progenitor cell stages, we identified the subset of genes in each stage-specific differentiation whose expression is directly altered by epigenetic modifications in their promoter sequences. Among the genes identified in the differentiation of neuro-epithelial and mid-radial glial cells, we note significant enrichment of genes implicated in schizophrenia, autism, bipolar, and depression (1.5–2.6 fold enrichment; Fisher's exact $p < 0.05$); a trend that becomes more pronounced when limited to genes implicated using genetic associations alone. With nearly 5–10% labelled as transcription factors (1.53–3.24 fold enrichment), this cross-disease enrichment provides a putatively causal mechanism early in neural development for the shared genetic correlation between these psychiatric phenotypes–an observation that is difficult to extract from GWAS data alone and is missed by differential peak/gene expression analyses.
(DOCX)

**S3 Text. We show, with a small extension to this *in silico* DNA reporter model to incorporate small molecule fingerprints, this network architecture can also be used to predict the transcriptomic impact of small molecules in cell lines, with as few as 600 molecules (24 hours post-exposure).** Predictive performance of this small molecule screen is on par with *in vitro* experimental replication of external test sets, allowing us to impute expression for all clinically approved molecules in the ChEMBL database. By training on the imputed expression profiles for all molecules first approved prior to the year 2000, we are able to retrospectively identify molecules that would later be assigned to the corresponding indications (post-2000), with 64–86% increase in F1-scores (i.e., the harmonic mean of precision and recall, a measure of model performance and accuracy that is robust to class imbalance), 22–63% increase in precision, 75–94% increase in recall, and 13–19% increase in area under precision-recall curve (AUPRC), compared to current chemoinformatic/molecule structure-based approaches.
(DOCX)

**S4 Text. List of URLs referenced in the main text, including links to peaBrain code, sample datasets, and Stage 1 scores.**
(DOCX)

## Author Contributions

**Conceptualization:** Moustafa Abdalla.

**Data curation:** Moustafa Abdalla.

**Formal analysis:** Moustafa Abdalla.

**Funding acquisition:** Moustafa Abdalla, Mohamed Abdalla.

**Investigation:** Moustafa Abdalla, Mohamed Abdalla.

**Methodology:** Moustafa Abdalla, Mohamed Abdalla.

**Project administration:** Moustafa Abdalla.

**Resources:** Moustafa Abdalla, Mohamed Abdalla.

**Software:** Moustafa Abdalla.

**Supervision:** Moustafa Abdalla, Mohamed Abdalla.

**Validation:** Moustafa Abdalla, Mohamed Abdalla.

**Visualization:** Moustafa Abdalla, Mohamed Abdalla.

**Writing – original draft:** Moustafa Abdalla.

**Writing – review & editing:** Moustafa Abdalla, Mohamed Abdalla.

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
