## [Decision Letter · Decision Letter 0]

11 Nov 2021

Dear Dr Abdalla,

Thank you very much for submitting your manuscript "A general framework for predicting the transcriptomic consequences of non-coding variation and small molecules" for consideration at PLOS Computational Biology.

As with all papers reviewed by the journal, your manuscript was reviewed by members of the editorial board and by several independent reviewers. In light of the reviews (below this email), we would like to invite the resubmission of a significantly-revised version that takes into account the reviewers' comments.

We cannot make any decision about publication until we have seen the revised manuscript and your response to the reviewers' comments. Your revised manuscript is also likely to be sent to reviewers for further evaluation.

Sincerely,

Eric Gamazon

Guest Editor

PLOS Computational Biology

Ilya Ioshikhes

Deputy Editor

PLOS Computational Biology

Reviewer's Responses to Questions

**Comments to the Authors:**

Reviewer #1: The authors tackle predicting transcriptional regulation at the tissue level to clarify the consequences of genetic variation. They develop a convolutional neural network framework, peaBrain, which they apply for a series of predictive tasks, such as identifying variants with a regulatory function or predicting allele-specific transcription factor binding sites. They show that peaBrain explains most of the variance in mean transcript abundance when DNA and tissue-specific annotations are employed. They also compare its performance with that of elastic net and deep-learning based frameworks, such as DeepSEA or MPRA.

The demonstrated predictive performance of peaBrian is convincing. The analysis steps are also extensively detailed and the choices involved are thoroughly justified. On the practical utility of the method: it seems that peaBrain can help reveal general insights, such as on the relevance of given tissues for a complex trait of interest, but it is less clear that it can provide finer resolution to the functional mechanisms involved. Nonetheless, the work tackles an important and timely topic, and the manuscript is well written.

I have the following additional comments/questions:

1. Current state of research: The field of deep learning for transcriptome prediction is rapidly evolving, but the references cited in the manuscript all date back to 2018 or earlier. The authors should revise the literature review presented in the introduction so it reflects the current state-of-the-art. They should also confirm and indicate why their proposal is still competitive in light of the most recent work.

2. Interpretability: To what extent finer level insights can be obtained from the output of peaBrain? It is well understood that machine learning approaches can have excellent predictive capacity yet this advantage often comes at the cost of interpretability due to their overparametrised nature. In particular, the comparison of peaBrain with other methods for the different tasks A-H lacks of insights as of to which feature(s) of the framework leads to its improved predictive performance. In this context, it seems difficult to completely rule out overfitting, although the cross-validation procedures used in peaBrain as well as the corroborative evidence from other studies and biological knowledge alleviate this concern. Comments on this would be helpful. Similarly, it would be informative to discuss why the competing methods perform poorly or fail to reach a performance comparable to that of peaBrain for the different tasks considered (e.g., lines 143-145). Such insights would provide an additional motivation for deploying the method in practice.

3. Tradeoff between computational efficiency and parameter choices: peaBrain relies heavily on cross validation and the authors repeatedly mention tradeoffs between computational tractability and accuracy. A runtime/memory profiling with respect to the sample size and for the different tasks A-H would be informative. This is particularly important as many recent studies involve samples sizes that are order of magnitude larger compared to GTEx. Also the choice of the 4 kbps core promoter window seems largely dictated by computational constraints, yet the performance is sensitive to this choice (lines 423-442). To alleviate this, it seems sensible to use tissue-specific windows; currently the optimisation of the window length was based on a single tissue (skeletal muscle). Would this improve predictive performance?

4. Data: GTEx constitute a rich resource for assessing the performance of peaBrain in a tissue-centric fashion. However it also requires careful considerations, in particular: (1) GTEx samples were collected post mortem which is prone to batch effects. Is there a way such effects can be diagnosed from the peaBrain output? The authors indicate that the RPKM values were adjusted for list of study parameters, which should alleviate the issue, yet this is at the expense of true biological variation being removed as well; (2) how sensitive is peaBrain to population stratification ? Here the first genotypes PCs were adjusted for, but it would nevertheless be interesting to evaluate the robustness of peaBrain to different degrees of stratification (or at least provide some discussion around how CNNs may be impacted by this issue); (3) the sample sizes for certain (less accessible) tissues can be rather small, e.g., for brain n < 250. In this case, the choice of k = 10 folds for the cross-validation scheme employed in peaBrain appears very large. For the elastic net model, the authors use k = 3 instead. What justifies these choices?

5. Transferability & deployment: The analyses presented in the manuscript suggest that the insights obtained with peaBrain may be relevant to a large panel of applications. The range of scenarios tested, including the additional experiments of Supplemental Notes 2 and 3 (page 16), suggests that the method is both transferable and extensible. The code is provided to the reader, however peaBrain involves a large number of parameter choices, whose specification likely require expert knowledge or, at the very least, extensive experience. For the method to be adopted by the community, it thus seems important to provide thorough guidance on the method’s settings under the form of a tutorial or a software package with dedicated help pages. Also, the framework relies on external annotations (tissue-specific or not) — for general annotations, could they be provided to the user or easily linked when the method is called by user?

6. Sharing information: Would peaBrain-like CNN architectures benefit from leveraging information across genes and/or tissues (rather than being applied separately for each gene/tissue) to exploit co-regulation patterns ? Would such set-up be hampered by computational constraints?

Reviewer #2: In this manuscript, Abdalla et al. described a tool (peaBrain) to predict transcriptomic consequences of non-coding variants and small molecules. I appreciate the endeavor for showing the general statistics about the comparison of their model with others and how the prediction is consistent with the general biological interpretation. However, I feel it lacks detailed examples about the exact non-coding variants (exact position, etc) and their effect on which diseases. For instance, how many predicted functional variants were validated by experiments in the literature. This is extremely important for attracting a wide spectrum of readers.

Some specific comments:

Figures need to be improved. Fonts are too small. The authors need to make figures more compact to save space.

Is peaBrain freely available for academic users? Can the authors provide detailed instruction for how to use peaBrain?

Reviewer #3: Moustafa et al. proposed a comprehensive framework for predicting the transcriptomic consequences. By incorporating the non-linearity feature from CNN, the authors successfully showed an insightful picture for gene expression regulation. The framework will be potentially very useful in multiple fields as the authors pointed out. However, details for the performance comparisons should be clarified to fully convince the readers.

Major comments

1. Line 46-47. Did the authors miss some earlier works (e.g., PMID 32433972) that performed cell-type-specific gene-expression predictions based on sequences? The Xpresso seems to be comparable with the step one of the proposed framework.

2. Line 86. A brief description of the “peaBrain variant” will be helpful. Did the dataset used to define the “peaBrain variant” overlap with the one used for eQTL analysis?

3. Line 142-143. (1) The regularized linear model is not quite comparable in this case.

4. The correlation between the “non-coding impact score” and the conservation score is interesting. If I understand correctly, the loci with high impact scores are not necessarily to be loci of common variants. Studies had been shown that the gene expression level of conserved genes tends to be less predictable by cis variants. The relationships among conservation, allele frequency, and expression predictability (or cis heritability) would be an interesting follow-up.

5. Line 277. The sample size should be mentioned given the p-value.

6. Line 330. The comparison between RTC and peaBrain is less comprehensive, compare with the quality in the other parts of the manuscript. Other tissue-prioritization methods need to be justified. e.g., PMID 27058395

7. Line 460. The best performing cis window size for expression abundance across genes and variance across individuals seems to be inconsistent (4k v.s. 1m).

8. Line 470. To my knowledge, the poor performance using elastic net is not consistent with the PrediXcan paper (Figure 3). More details should be provided. A r2 (elastic net) v.s. r2 (peaBrain) scatter plot would be very helpful to visualize to improvment. Ideally, training the prediction model in GTEx LCLs and estimating the performance in GEUVADIS LCLs.

9. A discussion for the very recent work from the DeepMind group will be welcome.

Minor comments

1. Line 55. Is Expecto computationally too expensive to have a larger cis window size?

2. Line 153. Type ‘OOS’.

3. Line 386. ‘F1-score’ should be defined for people do not familiar with the ML terminologies.

4. Line 416. The non-linear feature may also result from the activation function of CNN.

**Have the authors made all data and (if applicable) computational code underlying the findings in their manuscript fully available?**

Reviewer #1: Yes

Reviewer #2: None

Reviewer #3: None

PLOS authors have the option to publish the peer review history of their article (what does this mean?). If published, this will include your full peer review and any attached files.

Reviewer #1: No

Reviewer #2: No

Reviewer #3: **Yes: **Dan Zhou
---

## [Decision Letter · Decision Letter 1]

1 Mar 2022

Dear Dr Abdalla,

Thank you very much for submitting your manuscript "A general framework for predicting the transcriptomic consequences of non-coding variation and small molecules" for consideration at PLOS Computational Biology. As with all papers reviewed by the journal, your manuscript was reviewed by members of the editorial board and by several independent reviewers. The reviewers appreciated the attention to an important topic. Based on the reviews, we are likely to accept this manuscript for publication, providing that you modify the manuscript according to the review recommendations.

Sincerely,

Eric Gamazon

Guest Editor

PLOS Computational Biology

Ilya Ioshikhes

Deputy Editor

PLOS Computational Biology

[LINK]

Reviewer's Responses to Questions

**Comments to the Authors:**

Reviewer #1: The authors have put a lot of effort into improving the manuscript, and all my questions have been addressed. No further comment.

Reviewer #2: The authors have addressed my comments. I recommend the acceptance of the revised manuscript.

Reviewer #3: The authors addressed most of my concerns. However, I didn’t find adequate information for the comparison between elastic net and peaBrain in gene expression prediction across individuals.

The authors still need to detail the claim that only 28 of the 816 cis-heritable genes can be captured using elastic net. It would be great if the authors can show a transcriptome-wide picture for the predictability comparison between peaBrain and elastic net when the sample size varies.

**Have the authors made all data and (if applicable) computational code underlying the findings in their manuscript fully available?**

Reviewer #1: Yes

Reviewer #2: None

Reviewer #3: None

PLOS authors have the option to publish the peer review history of their article (what does this mean?). If published, this will include your full peer review and any attached files.

Reviewer #1: No

Reviewer #2: No

Reviewer #3: No

Figure Files:

Data Requirements:

Reproducibility:

References:

---

## [Editor Report · Decision Letter 2]

16 Mar 2022

Dear Dr Abdalla,

We are pleased to inform you that your manuscript 'A general framework for predicting the transcriptomic consequences of non-coding variation and small molecules' has been provisionally accepted for publication in PLOS Computational Biology.

Best regards,

Eric Gamazon

Guest Editor

PLOS Computational Biology

Ilya Ioshikhes

Deputy Editor

PLOS Computational Biology

---

## [Editor Report · Acceptance letter]

11 Apr 2022

PCOMPBIOL-D-21-01433R2 

A general framework for predicting the transcriptomic consequences of non-coding variation and small molecules

Dear Dr Abdalla,

I am pleased to inform you that your manuscript has been formally accepted for publication in PLOS Computational Biology. Your manuscript is now with our production department and you will be notified of the publication date in due course.

With kind regards,

Agnes Pap
